# Implementation of a marketing plan for the dissemination of the WHO SkinNTDs app in Cameroon

Henri Claude Moungui[1,2]*, Paul Tonkoung Iyawa[3], Hugues Nana-Djeunga[2], Jose Antonio Ruiz-Postigo[4], Carme Carrion[1,5]

1 Universitat Oberta de Catalunya, Barcelona, Spain, 2 Higher Institute for Scientific and Medical Research, Yaounde, Cameroon, 3 Helen Keller International, Maroua, Cameroon, 4 Prevention, Treatment and Care Unit, Department of Control of Neglected Tropical Diseases, World Health Organization, Geneva, Switzerland, 5 eHealth Lab Research Group, eHealth Center and School of Health Sciences, Universitat Oberta de Catalunya, Barcelona, Spain

* henrimoungui@yahoo.fr

## Abstract

### Background

Skin-related neglected tropical diseases (sNTDs) remain a significant public health challenge in Cameroon, where limited resources, training, and infrastructure hinder early diagnosis and management. The World Health Organization (WHO) has developed the SkinNTDs app version 3.0 as a digital solution to assist frontline healthcare workers (FHWs) in recognizing and managing sNTDs. As utilization will rely on a high degree of awareness among FHWs, a dedicated and effective marketing plan is required. This study describes the design, implementation, and evaluation of a structured marketing plan to promote the app among FHWs in Cameroon.

### Methods

We conducted a pilot quasi-experimental before-and-after study comparing three 6-month phases—pre-campaign, campaign, and post-campaign. The multi-channel marketing campaign combined communications via WhatsApp, in-person training sessions, video presentations, and email outreach. Google Play Console Analytics provided monthly metrics on store listing visits, downloads, installations, uninstalls, and retention.

### Results

During the campaign (1 April 2024–30 September 2024), store page visits totaled 961, yielding 616 downloads (conversion rate = 64.1%) and the app was installed 751 times; new-user acquisition exceeded 81.6%. Net installs surged by 227.3% in April and 140.4% in May, with retention above 88%. ANOVA revealed significant period

**Data availability statement:** All relevant data are within the paper and its Supporting Information files.

**Funding:** The author(s) received no specific funding for this work.

**Competing interests:** The authors have declared that no competing interests exist.

effects on growth rate (p = 0.007, $\varepsilon^2 = 0.591$), loss rate (p = 0.011, $\eta^2 = 0.452$), churn rate (p = 0.003, $\eta^2 = 0.532$), and retention rate (p = 0.003, $\eta^2 = 0.532$), with campaign performance superior to pre- and post-campaign phases. Interrupted time series analyses found gradual adoption and sustained retention following intervention start, but significant decrease at campaign end.

## Conclusion

A context-adapted, multi-channel marketing strategy markedly improved adoption and retention of the WHO SkinNTDs app among Cameroonian FHWs. Digital (WhatsApp, videos) and face-to-face (training) channels were complementary. Sustained integration into routine health activities and automated re-engagement tools are recommended to maintain long-term use and inform scale-up in other endemic settings.

## Introduction

### Skin-related neglected tropical diseases

Skin-related neglected tropical diseases (sNTDs) include Buruli ulcer, cutaneous leishmaniasis, post-kala-azar dermal leishmaniasis, leprosy, lymphatic filariasis (lymphedema and hydrocele), mycetoma and other implantation mycoses (e.g., chromoblastomycosis and sporotrichosis), noma, onchocerciasis (river blindness), scabies, tungiasis and yaws [1]. Many of the resource-limited countries where these diseases are prevalent face significant challenges to capacity building for diagnosis and surveillance [2]. Isolated and often inadequately trained frontline health workers (FHWs) struggle to provide more than the most basic care, and a lack of integrated surveillance systems and diagnostic tools can hinder the early identification and treatment of sNTDs [3]. Thus, there is an urgent need for innovative capacity-building solutions to improve the detection and clinical management of sNTDs in resource-limited settings.

### The role of mHealth in sNTDs

Digital health tools can support FHWs in the diagnosis of sNTDs through remote reviews and consultations with patients [4]. A systematic review of apps available through Google Play or the Apple App Store identified up to fifteen apps that addressed sNTDs, most of which targeted health care professionals [5]. Among the few that focused on an integrated approach to the various diseases was the World Health Organization (WHO) SkinNTDs app.

The WHO SkinNTDs app was developed from a training guide for FHWs in recognizing the signs and symptoms of NTDs affecting the skin [6,7]. Version 3.0 (v. 3.0.0) is the result of merging version 2 with the SkinApp developed by Until No Leprosy Remains, which encompassed information on 24 common skin diseases [8,9]. Field evaluations of the WHO SkinNTDs app in Ghana and Kenya [10], and Cameroon [11] concluded that the app moderately meets the needs of users as a capacity building tool.

The app is freely available for download and use on both android and iOS devices from Google Play and Apple App stores. Once a user has opened the app it will function offline. The welcome menu displays the functionalities of the app: 'Signs & symptoms', 'Diagnoses', 'How to manage', 'Glossary' and 'Skin NTDs Learning hub'. Examples of screenshots from Skin NTDs App are available on Internet [12,13].

## Problem statement

The potential of the WHO SkinNTDs app to transform sNTD management is limited by a lack of both awareness about its existence and the training needed for effective utilization among FHWs in endemic regions. As yet, there are no official guidelines for effective dissemination of the app.

To address these challenges, a structured marketing plan is critical to promote it, ensure widespread adoption, and encourage long-term use of the app among FHWs. The plan should include strategies to raise awareness of the app, and provide training for its use. It should also address such barriers to implementation as effects on the patient-provider interaction and any misalignment with personal values, and communicate how the app mitigates limited internet access [14–16].

The published literature suggests that successful adoption depends on final users perceiving a technology as possessing a high level of both usefulness and ease of use [5,16]. The trustworthiness of the brand promoting the app and of the information presented in the app have also been found to be factors in the adoption of mHealth apps [17]. Yet, health professionals will not use reputable and effective mHealth apps of which they are not aware [18].

As noted in a recent scoping review, there is little information in published studies on exactly how individuals became "adapted" to the use of an app over time [4]. We hypothesize that a well-designed and fully documented marketing plan for the WHO SkinNTDs app, and its systematic implementation, will provide valuable information on this issue. We argue that such a plan would require accurate definitions of the target end users, the most effective communication channels and advertising content. It would also necessitate an appropriate and achievable timeline for reaching and motivating end users to adopt and maintain engagement with the app.

## Objective of the paper

This paper describes the development, implementation, and evaluation of a marketing plan for the WHO SkinNTDs app in the resource-limited country of Cameroon.

Specifically, the paper:

1. Outlines the process of creating a marketing plan.

2. Provides details of how the plan was implemented, including the strategies used to raise awareness, train health workers, and promote app usage.

3. Analyzes the effectiveness of this implementation using metrics on app downloads, usage, and user feedback.

4. Provides a critical analysis of the implementation process that identifies successful aspects, challenges, and the knowledge gained.

5. Offers policy implications and practical recommendations for scaling up app use in other low- and middle- income countries (LMICs) where sNTDs are endemic.

By addressing these objectives, this paper contributes to the growing body of evidence-based literature on mHealth interventions for sNTDs and provides a roadmap for the successful dissemination of similar tools in resource-limited settings.

## Materials and methods

### Study settings

This pilot study was implemented in Cameroon, a LMIC in Central Africa. The country has a pyramidal health system structured in three levels: a central level supervised by the Ministry of Public Health, which defines public health strategies; an intermediate level that coordinates regional activities; and a peripheral level of health districts and health areas responsible for implementing policies [19]. Although health districts in the North and Far North regions were initially targeted in this study, awareness of the app has reached other areas through word of mouth and social media diffusion via WhatsApp and LinkedIn.

Cameroon is endemic for multiple neglected tropical diseases (NTDs), with the Far North and North regions particularly affected by onchocerciasis, lymphatic filariasis, schistosomiasis, soil-transmitted helminthiasis, and trachoma [19]. The Far North region has recently witnessed a resurgence of leishmaniasis, yaws, and Guinea worm, as well as the emergence of noma. Thus, the country is in urgent need of improved case management and diagnostic tools [20,21].

The National NTD Control Program mirrors the health system's pyramidal structure. At the central level, a coordination unit, housed within the sub-directorate for NTDs and Malaria, oversees thematic programs such as the Onchocerciasis Control Program, and the Program for the Fight against Leprosy, Leishmaniasis, Buruli Ulcer, and Yaws [19]. Regional focal points coordinate activities at the intermediate level, while health personnel and community health workers implement interventions at the peripheral level. However, dermatological care remains underdeveloped at the peripheral level, as the few specialists in the country are concentrated in urban centers, leaving generalist health personnel to manage skin conditions in rural areas [22].

As stated in the National Digital Health Strategic Plan 2020–2024 [15], only 32.1% of health facilities in Cameroon have their own computers, with an additional 16.8% using private computers, and internet access is limited to 27% of health facilities. Private mobile phones purchased by staff members, but used by the facility, are the most common means of communication (52.7%). Then follow mobile phones belonging to the facility (33.2%), and private mobile phones with airtime paid for by the facility (23.1%). Moreover, remote health facilities often lack electricity and internet access or have inadequate and unreliable supplies.

### Theoretical framework

The study's dissemination strategy drew on two behavioral frameworks: the Diffusion of Innovations (DOI) theory, and the Technology Acceptance Model (TAM) [23]. DOI guided the campaign's structure, including phased rollout (launch, engagement, embedment) and channel selection, aligning with its constructs of relative advantage and social system diffusion. TAM informed messaging to highlight perceived usefulness (e.g., diagnostic accuracy) and ease of use (e.g., offline functionality, ease of download/installation), targeting FHWs' behavioral intent. A/B testing [24] of materials (e.g., video formats) further optimized compatibility and perceived value, reinforcing theoretical grounding.

### Development of the marketing plan

A marketing plan is a strategic summary that demonstrates an understanding of the product, audience, message, and value, and can be an indicator of how successful an app will be when launched [24]. Typical components of a not-for-profit mHealth app marketing plan include marketing goals and objectives, product objectives, market analysis (population targeted by the app), competition (leading apps with similar aims), a strengths-weaknesses-opportunities-threats (SWOT) analysis of building the app, app functionality (including special features that give the app a competitive edge), and a timeframe [25]. To develop the marketing plan, we began by conducting a systematic review of dissemination strategies for mHealth apps.

Our review identified the key strategies commonly used for disseminating mHealth apps. These include social media campaigns, email marketing, and face-to-face communication to drive downloads, alongside animated graphics and short videos to boost engagement. Techniques such as push notifications and incentivization have been shown to sustain user engagement, while effectiveness can be assessed through such metrics as downloads, adherence rates, and cost per click [26].

From this list of dissemination strategies, we then selected those that would best suit the Cameroonian context. From our knowledge of the Cameroonian health system and ICT infrastructure, we deduced that WhatsApp, printed posters, and face-to-face communication would be the most appropriate channels for communication with FHWs, with videos and presentations about the app shared with strategic level stakeholders through email. To start drafting the plan, we took inspiration from the steps described in the "Marketing Your Telehealth Program" toolkit [27].

The target market for the WHO SkinNTDs app is the national health system, particularly health facilities in NTD-endemic areas. The primary audience for dissemination messages is FHWs in these regions. Both virtual and face-to-face opportunities would be used to reach the audience and promote the app.

Dissemination messages for the WHO SkinNTDs app were developed through a structured process that included:

1. Identifying key functionalities: Messages highlighted app features, such as 'Signs and Symptoms' to simplify sNTDs diagnosis.

2. Targeting FHWs: Messages were tailored to address the needs of FHWs in sNTD-endemic areas by emphasizing how the app could enhance their daily practice.

3. A/B Testing: Alternative versions of promotional content, especially videos, were tested to determine the most effective format.

4. Multi-channel approach: Messages were adapted for various platforms, including WhatsApp, videos, and in-person training, to ensure accessibility and broaden reach.

5. Local adaptation: Content addressed local challenges, such as limited internet access, and emphasized the practical usability of the app.

6. Information and brand trustworthiness: Messages emphasized that information provided in the app has been developed by the globally-renowned WHO, thereby increasing trustworthiness. Furthermore, the app features the WHO logo. It was also explained that the team received ethical clearance and administrative authorizations.

### Dissemination materials refinement process

We piloted an A/B testing process to refine the main dissemination video. Ten purposefully selected stakeholders from central, regional, and district levels viewed the instructional Video A via WhatsApp and provided feedback on clarity, length, and suggested improvements. Responses were anonymized, thematically coded, and analyzed in NVivo 12 Plus. Video B incorporated these refinements. A detailed description of the iterative refinement process is provided in supporting information S1 Appendix.

Separately, an introductory PowerPoint was converted to PDF and shared via WhatsApp after training sessions, containing the app's background, objectives, download link, QR code, and clearance details to build trust. Clearance details were later omitted, and the lead field investigator made informal, iterative adjustments during the campaign to suit audience needs and key messages. Materials were developed in French/English versions and adapted to the audience throughout the campaign.

### Study design

We piloted a quasi-experimental, before-and-after study to evaluate the impact of a structured, multi-channel marketing intervention on the adoption and retention of the WHO SkinNTDs app among frontline health workers (FHWs) in

Cameroon. The intervention phase took place from April 11 to September 30, 2024, during which targeted participants were actively enrolled through field activities.

FHWs were identified and invited to participate in the campaign in convenience sampling within target districts. The reason behind was to adapt with operational feasibility and to stick with the idea of integration into the health system. Thus, the research team was constantly looking for opportunities when programs in the ministry of public health organized training sessions or meeting with FHWs. The marketing campaign included coordinated WhatsApp broadcasts, in-person training workshops, instructional videos, and targeted email outreach.

Prior to this campaign, there was no documented dissemination strategy in place, and the campaign was discontinued after September 30, 2024. However, the app had been publicly available for download, both in Cameroon and globally, at least six months before the campaign began and remained accessible afterward. The evaluation therefore covered three consecutive six-month periods: the pre-campaign phase (1 October 2023 to 31 March 2024), the campaign phase (1 April 2024 to 30 September 2024), and the post-campaign phase (1 October 2024 to 31 March 2025).

## Data collection and summary statistics

The collection and analysis method complied with the terms and conditions for the source of the data. Target users were invited to download the app from the Google Play Store. In Cameroon, Android and iOS are the dominant mobile operating systems, with Android holding 79–84% of the market share during the 18-months study period [28]. Given this distribution, the study focused exclusively on Android users. Usage data were automatically captured via Google Play Console, a platform used by developers to publish and monitor app performance. Google Play Console provided aggregated metrics related to how users discovered the app, when they downloaded it, the devices used, app launch events, and uninstall activity. These data allowed us to evaluate the effectiveness of the marketing campaign and to identify potential areas for improvement. Definitions of all metrics, as given in Google Play Console Help [29], are provided in supporting information file S2 Appendix.

Data extracted covered the 18-month period including pre- and post-campaign. Pre-campaign data were already available before the intervention began, while campaign and post-campaign data were collected prospectively without any direct involvement from the research team.

All metrics were obtained in compliance with Google's Terms of Service (https://play.google.com/intl/en/about/play-terms). Data were anonymized and aggregated by Google, and no personally identifiable information was accessible to the research team.

We first aggregated the Google Play Console data by months. These aggregated values were presented in summary tables and graphics to provide a clear overview of trends over time.

## Inferential statistics

To assess whether performance metrics differed across the three periods, we conducted analyses of variance (ANOVA) for outcomes meeting the assumptions of normality (Shapiro–Wilk test) and homogeneity of variance (Levene's test). When either assumption was violated, we applied the nonparametric Kruskal–Wallis test instead. Significant omnibus results were followed by pairwise comparisons: Tukey's Honestly Significant Difference (HSD) test for ANOVA when assumptions were met, and the Dwass–Steel–Critchlow–Fligner (DSCF) test for Kruskal–Wallis outcomes when assumptions were not met [30]. Effect sizes were reported to indicate the practical importance of differences, using Cohen's d (with 95% confidence intervals) for t-tests, eta-squared ($\eta^2$) for ANOVA, and epsilon-squared ($\varepsilon^2$) for Kruskal–Wallis tests. Statistical significance was set at $\alpha = 0.05$. While Tukey's HSD and DSCF tests controlled the Type I error rate for pairwise comparisons within each outcome, no statistical correction for multiple testing (e.g., Bonferroni, Holm, or False Discovery Rate – FDR) [31] was applied across the six primary outcomes (store listing conversion, device conversion, growth, loss, churn, and retention rates). Analyses were conducted using Microsoft Excel and Jamovi for Windows (version 2.6.44) [32].

### Interrupted time series analyses

The quasi-experimental nature of our study calls for appropriate statistical methods to uncover the potential effects of an intervention over the time. Interrupted time series (ITS) analysis is a valuable study design for evaluating the effectiveness of population-level health interventions that have been implemented at a clearly defined point in time [33]. It is the next best approach when, like in our study, randomization is not possible [34]. There are many ITS analytical tools available [35], and their choice depends on the nature of both data and the intervention to evaluate.

We conducted three ITS analyses to evaluate the impact of the SkinNTDs app dissemination campaign in Cameroon. First, only for Cameroon, an auto regressive integrated moving average (ARIMA) error structure (ARIMA(2,0,0)) modeled both level and slope changes in Install base during and post-campaign while accounting for autocorrelation [36]. Second, a generalized additive mixed model (GAMM) compared Cameroon to Nigeria, Ghana, and Ethiopia, to flexibly model nonlinear pre/post-intervention trends in Install base while controlling for country-level heterogeneity [35]. Third, in another cross-country comparison (Cameroon vs Nigeria, Ghana, Ethiopia), a Poisson generalized linear mixed-effects model (GLMM) assessed Daily active devices (DADs) with country random effects and interaction terms for intervention effects. Install base metric (30-day rolling number of active devices on which the app is installed) and Daily active devices (Average number of active devices that open the app each day) were chosen as the best proxy measures of both installations and retention while accounting for sideloads (installations of the app on Android devices external to Google Play store). Being all endemic to NTDs, the three countries were selected because they were top African countries with highest install base average values during the 18-months period. These countries also had similar mobile operating system market share with Cameroon during that period [28].

Model selection used Akaike/Bayesian Information Criteria (AIC/BIC). Residual diagnostics included Ljung-Box tests, ACF plots, and Q-Q normality checks and all these analyses were performed in R 4.5.0 with $\alpha = 0.05$ for significance threshold. Full descriptions of ITS procedures including model specifications are available in supporting information file S3 Appendix.

### Ethics approval

This study received two ethical approvals: from the Universitat Oberta de Catalunya, Barcelona, Spain (CE22-TE29, signed on April 26, 2022); and from the National Ethics Committee for Health Research in Cameroon (N°2024/01/1620/CE/CNERSH/SP, signed on January 17, 2024). In addition, this study received administrative research authorization from the directorate of disease control and the regional delegates of public health in the Far North and North regions in Cameroon.

All participants were provided with information about the study and it was explained that they could withdraw at any time without prejudice. Each participant provided oral informed consent and all data used for analysis were anonymized.

### Inclusivity in global research

Additional information regarding the ethical, cultural, and scientific considerations specific to inclusivity in global research is included in the supporting information file S1 Checklist.

## Results

### Marketing plan developed for the WHO SkinNTDs app in Cameroon

The plan encompassed the objectives of raising awareness, driving downloads, and fostering sustained use of the app. It was structured into three phases: launch, engagement, and embedment. A multi-channel approach was employed, leveraging digital and in-person strategies tailored to the Cameroonian context. Below is a summary of the plan. The full version is available in supporting information file S4 Appendix.

1. **Target audience**

The primary users were FHWs (doctors, nurses, and assistant nurses) in urban, semi-urban, and rural areas, particularly in regions endemic for sNTDs. These settings often experience limited internet access, electricity shortages, and scarce dermatological expertise.

2. **Core objectives**

- Increase awareness: Reach ≥80% of target FHWs in North/Far-North Cameroon via WhatsApp, emails, or in-person sessions within 6 months.

- Training coverage: Conduct ≥20 in-person sessions.

- Drive downloads: Achieve ≥60% conversion rate from store visits to downloads.

- Sustain engagement: Maintain ≥80% user retention 3 months post-installation.

3. **Communication channels**:

- Digital: WhatsApp, email, and video presentations (e.g., short promotional videos with A/B testing for messaging).

- In-Person: Training sessions, PowerPoint presentations with embedded QR codes, and ad-hoc interactions during health campaigns.

- Print and media: Advertisements in local health journals and TV broadcasting in collaboration with the Ministry of Public Health.

4. **Messaging strategy**

- Emphasize app utility (e.g., offline functionality, WHO credibility) and ease of use.

- Highlight benefits for routine practice, such as improving diagnostic accuracy and case management while serving as a memory aid for capacity building.

- Include calls-to-action (e.g., direct download links, QR codes).

5. **Phased implementation**

- Launching (January-February 2024): Official announcements via email, a one-day hybrid meeting with stakeholders, and dissemination of promotional materials.

- Engagement (February-June 2024): WhatsApp group activations, push notifications, and follow-up training sessions.

- Embedment (June 2024): Integration into health system activities (e.g., NTD campaigns) to institutionalize app use.

6. **SWOT analysis**

- Strengths: WHO branding, multilingual support, user-friendly design, offline functionality once installed.

- Weaknesses: Research conducted in Ghana, Kenya, and Cameroon identified a need for greater customization, engagement, and interactivity in the app [10,11].

- Opportunities: There is global momentum for sNTD elimination [1,22] and synergies with tele dermatology apps (e.g., eSkinHealth [37]).

- Threats: Low internet coverage in remote areas and voluntary uninstalls

## Dissemination activities implemented

Communications started on April 6, 2024. Initial contacts (emails, WhatsApp, phone calls) were made with Ministry of Public Health staff at central and regional levels in order to establish a hierarchical network chain and reach personnel at lower levels. Field activities took place between April 11, 2024, and September 30, 2024. These started with practical presentations at regional and district levels. While continuously looking for every occasion when FHWs would assemble, we attempted to arrange a presentation upon confirmation. When successful, we adapted our approach depending on the audience, the location, the type of activity that had prompted the meeting, and the time allocated, which varied between 10 and 30 minutes. We staged a PowerPoint presentation or a practical demonstration, or a combination of the two.

As a steep decrease in downloads and active users was noted in June, we hired two research assistants to raise awareness about the app through in-person presentations at ten health facilities each in the North and Far North regions. Most of the participants at these sessions were able to download and install the app on their devices. Whenever possible, attendance sheets were collected and each participant contacted for a follow-up on WhatsApp, during which they were given a PDF copy or video presentation about the app, or the download link.

The video presentation opens by discussing the rationale behind the app and highlighting the challenges to capacity building for early sNTDs diagnosis, then explains how the app was developed by WHO, how it works, and how it can be navigated. The videos were produced in French and English, the two official languages of Cameroon. The video narrative is available in supporting information file S1 Appendix.

Research team members enrolled participants in a WhatsApp group specifically created to promote the app following a training session. In total, 57 individuals were enrolled in the group, but two had left by September 30, 2024. During the campaign, 227 messages were exchanged in the WhatsApp group. Message frequency peaked on Mondays (n=51), with the highest activity occurring in September 2024 (n=113) and July 2024 (n=79). Most messages were sent during morning (8:00–10:59 AM) and evening (5:00–8:59 PM) periods. The primary author was the most active participant, and the Google Play Store link to the app was shared six times in the group (three times each in May and July 2024). Additionally, 98 media files—including instructional videos, PDFs presentation of the app, screenshots from the app, and field photos of suspect cases—were exchanged. Beyond the group, the primary author promoted the app individually, sharing, for example, the Play Store link approximately 50 times in private chats. The research team also shared information about the app in various other groups used by the Ministry of Public Health regional and district services. Dissemination activities executed are summarized in Table 1.

## Description of messages shared

During practical sessions, we communicated the following information: how the idea for the app originated, who developed the app, and how we are working to disseminate the app We also explained how to get the app, including how to search for the app and identify it in Google Play store through its name and the WHO logo, and how to download and install it. Next, we explained how to select the desired language when initially launching the app, how to navigate within the app and the features available. Messages were tailored to address factors known to positively influence uptake and engagement with mHealth apps, such as perceived utility [38], free availability [39], and educational information/material [40]. We also communicated that once the user has downloaded, installed, and selected interface language, that version of the app can be used without an internet connection. This was a particularly critical point as the remote areas FHWs work in often lack reliable electricity and internet services [15], which can hinder the implementation of mHealth apps [18,41] and health management information systems in general [42].

## Results of video refinement process

Overall, participants found Video A clear, understandable, and informative, with several describing it as "easy to follow". Some, however said that the video "requires strong background knowledge on NTDs to follow". Suggestions for

**Table 1. Implemented activities.**

| What | Where | Who | When |
|------|-------|-----|------|
| 12 in-person practical presentation sessions | • **03** state registered nurse schools.<br>• **05** district offices.<br>• **04** meeting/training halls for activities organized by other health programs. | **412 participants:**<br>Third-year nurse students, health workers, NGO staff. | **April 11 to September 20, 2024** |
| **One-to-one demonstrations** | Any encounters with individuals. | **Approximately 100 health workers.** | **April 4 to September 30, 2024** |
| **Activation of a WhatsApp group to foster continuous use of the app, provide user support and foster disease surveillance** | WhatsApp. | **55 health workers.** | **Since May 6, 2024** |
| **Diffusion of a video presentation** of the WHO SkinNTDs app (first version) | WhatsApp and LinkedIn. | **>800 members.** | **June 15, 2024** |
| **Diffusion of a video presentation** of the WHO SkinNTDs app (second version) | WhatsApp and LinkedIn. | **>800 members.** | **July 1, 2024** |
| **18 in-person practical presentation sessions** | •**18** health facilities. | **Approximately 100 health workers.** | **July 4 to −20, 2024** |

improvement included slowing transitions between images to allow better reading of disease names, enlarging and zooming in on lesion areas during explanations, and producing additional tutorials to guide users through specific app functions such as making a diagnosis. A few also recommended extending the video length to provide more detail. Full responses collected are given in supporting information file S1 Appendix.

## Results of the dissemination activities

**Summary statistics.** The effectiveness of the dissemination campaign was evaluated using metrics derived from Google Play Console data. Monthly aggregates for store listing metrics and user acquisition parameters are summarized in Table 2.

During the campaign period (April-September 2024), the app's store page received 961 visits leading to 616 downloads (conversion rate = 64.1%). In total, 731 users installed the app. The store listing conversion rates remained robust, ranging from 58.4% to 70.8% for all users and 56.7% to 71.3% for new users.

The high proportions of new visitors (≥89.7%) and new-user acquisitions (≥81.6%) demonstrate a strong outreach among previously untapped audiences within the target population. The user-to-device ratio remained near 1.0, indicating that most visitors acquired the app on a single device. Installation dynamics, installed audience growth, and retention performances are reported in Table 3.

The app was installed 751 times and uninstalled 104 times during the campaign. Thus, the app remained installed on 647 devices (net installs) by September 30, 2024 and still on 594 devices by March 31, 2025. Overall, the campaign was marked by an increase in the number of install events and net installs during the campaign period. For example, April 2024 witnessed 104 install events with a net increment of 101 installs, while May 2024 reached 232 install events and 216 net installs. The growth rate during the campaign peaked dramatically (e.g., 227.3% in April and 140.4% in May), suggesting an immediate impact of dissemination activities.

Loss rate and churn rate values remained low during the campaign (loss rates ranged between 2.6% and 8.1%, and churn rates between 5.9% and 11.3%), thereby contributing to overall high user base retention rates, which generally remained above 88%. In contrast, dynamics were less favorable in both the pre-campaign (1 October 2023–31 March 2024) and post-campaign (1 October 2024–31 March 2025) periods, with lower and even negative net growth and relatively higher churn rates, underscoring the efficacy of the campaign.

Table 2. Store listing metrics and user acquisition.

| Months | Sum of store listing visitors | Sum of store listing acquisitions | Sum of user acquisition | Store listing conversion rate | Store listing conversion rate – new users | Users to devices ratio | Proportion of new store listing visitors | Proportion of acquisitions from new users |
|---|---|---|---|---|---|---|---|---|
| Oct-23 | 6 | 4 | 10 | 66.7% | 60.0% | 1.1 | 83.3% | 50.0% |
| Nov-23 | 5 | 4 | 7 | 80.0% | 75.0% | 1 | 80.0% | 42.9% |
| Dec-23 | 3 | 2 | 5 | 66.7% | 66.7% | 1 | 100.0% | 60.0% |
| Jan-24 | 3 | 2 | 5 | 66.7% | 50.0% | 1 | 66.7% | 20.0% |
| Feb-24 | 8 | 0 | 2 | 0.0% | 0.0% | 1.0 | 100.0% | 0.0% |
| Mar-24 | 8 | 4 | 0 | 50.0% | 20.0% | 0.0 | 62.5% | – |
| Apr-24 | 139 | 94 | 104 | 67.6% | 67.4% | 1.0 | 92.8% | 88.5% |
| May-24 | 306 | 190 | 229 | 62.1% | 62.0% | 1.0 | 99.7% | 97.8% |
| Jun-24 | 101 | 59 | 66 | 58.4% | 56.7% | 0.8 | 96.0% | 92.4% |
| Jul-24 | 251 | 164 | 199 | 65.3% | 64.2% | 1.0 | 96.8% | 92.0% |
| Aug-24 | 68 | 41 | 49 | 60.3% | 59.0% | 0.8 | 89.7% | 81.6% |
| Sep-24 | 96 | 68 | 84 | 70.8% | 71.3% | 0.9 | 97.9% | 94.0% |
| Oct-24 | 16 | 8 | 18 | 50.0% | 46.7% | 0.7 | 93.8% | 50.0% |
| Nov-24 | 7 | 4 | 14 | 57.1% | 57.1% | 0.6 | 100.0% | 35.7% |
| Dec-24 | 11 | 5 | 12 | 45.5% | 45.5% | 0.5 | 100.0% | 50.0% |
| Jan-25 | 9 | 5 | 14 | 55.6% | 33.3% | 0.7 | 66.7% | 21.4% |
| Feb-25 | 15 | 10 | 18 | 66.7% | 66.7% | 0.8 | 100.0% | 61.1% |
| Mar-25 | 12 | 7 | 25 | 58.3% | 58.3% | 0.8 | 100.0% | 28.0% |
| Grand total | 1064 | 671 | 861 | 63.1% | 62.1% | 0.9 | 96.1% | 85.0% |
| Campaign total | 961 | 616 | 731 | 64.1% | 63.5% | 0.9 | 96.7% | 92.9% |

Monthly trends in app user onboarding are depicted in Fig 1, and the app user retention trends are shown in Fig 2. A focused view of the retention dynamics (growth rate vs churn rate) during the campaign period is provided in Fig 3. Collectively, these figures suggest a considerable effect of our dissemination activities, which resulted in rapid improvements in user acquisition, growth and retention dynamics during the active campaign period, and a subsequent decline.

## Statistical analysis of performance metrics by campaign period

Assumption checks indicated that normality and/or homogeneity of variance were violated for most metrics, except loss rate, churn rate, and user base retention. For store listing conversion rate (all users, new users, returning users), Kruskal–Wallis tests revealed no significant differences across periods (all $p \geq 0.310$, $\varepsilon^2 \leq 0.084$), with Dwass–Steel–Critchlow–Fligner (DSCF) pairwise comparisons confirming the absence of significant contrasts.

For device conversion rate (all users), the Kruskal–Wallis test detected a significant difference across periods ($\chi^2 = 9.28$, $df = 2$, $p = 0.010$, $\varepsilon^2 = 0.045$). DSCF post hoc tests showed higher conversion in the post-campaign period compared with both the pre-campaign ($p = 0.024$) and campaign ($p = 0.038$) periods. Growth rate differed significantly across periods ($\chi^2 = 10.0$, $df = 2$, $p = 0.007$, $\varepsilon^2 = 0.591$), with DSCF tests indicating higher values during the campaign compared to both post-campaign ($p = 0.011$) and pre-campaign ($p = 0.043$) periods.

For loss rate, ANOVA results ($F(2,15) = 6.18$, $p = 0.011$, $\eta^2 = 0.452$) with Tukey's HSD adjustment showed significantly lower rates in the campaign ($p = 0.012$, Cohen's $d = -1.912$, 95% CI [−3.35, −0.47]) and post-campaign ($p = 0.043$, $d = -1.546$, 95% CI [−2.92, −0.17]) periods compared to pre-campaign. Churn rate also differed significantly ($F(2,15) = 8.53$,

**Table 3.  Installation, growth, and retention metrics.**

| Months | Sum of install events | Average daily active devices | Sum of uninstall events | Installed audience - first day of the month | Installed audience - last day of the month | Net installs | Growth rate | Loss rate | Churn rate | User base retention rate |
|---|---|---|---|---|---|---|---|---|---|---|
| Oct-23 | 7 | 1.5 | 5 | 65 | 67 | 2 | 4.6% | 10.4% | 10.8% | 89.2% |
| Nov-23 | 6 | 1.8 | 3 | 65 | 60 | 3 | 3.1% | 8.3% | 7.7% | 92.3% |
| Dec-23 | 3 | 1.3 | 2 | 66 | 61 | 1 | −1.5% | 9.8% | 9.1% | 90.9% |
| Jan-24 | 3 | 1.7 | 0 | 59 | 58 | 3 | 1.7% | 6.9% | 6.8% | 93.2% |
| Feb-24 | 1 | 1.4 | 3 | 60 | 52 | −2 | −8.3% | 13.5% | 11.7% | 88.3% |
| Mar-24 | 3 | 1.2 | 6 | 49 | 44 | −3 | −11.5% | 13.6% | 11.5% | 88.5% |
| Apr-24 | 104 | 7.1 | 3 | 44 | 151 | 101 | 227.3% | 2.6% | 9.1% | 90.9% |
| May-24 | 232 | 13.0 | 16 | 155 | 393 | 216 | 140.4% | 4.3% | 11.3% | 88.7% |
| Jun-24 | 78 | 8.1 | 14 | 392 | 439 | 64 | 10.9% | 5.2% | 5.9% | 94.1% |
| Jul-24 | 200 | 19.0 | 33 | 431 | 590 | 167 | 34.4% | 8.1% | 10.9% | 89.1% |
| Aug-24 | 52 | 10.6 | 24 | 600 | 594 | 28 | 0.8% | 7.4% | 7.5% | 92.5% |
| Sep-24 | 85 | 11.0 | 14 | 599 | 628 | 71 | 8.1% | 5.7% | 6.1% | 93.9% |
| Oct-24 | 13 | 6.7 | 18 | 631 | 601 | −5 | −2.9% | 6.0% | 5.7% | 94.3% |
| Nov-24 | 9 | 6.0 | 21 | 620 | 557 | −12 | −5.5% | 8.4% | 7.8% | 92.2% |
| Dec-24 | 11 | 4.6 | 18 | 568 | 536 | −7 | −4.7% | 7.1% | 6.8% | 93.2% |
| Jan-25 | 8 | 5.5 | 12 | 514 | 498 | −4 | −3.0% | 6.0% | 5.6% | 94.4% |
| Feb-25 | 16 | 4.7 | 10 | 506 | 497 | 6 | −1.8% | 5.4% | 5.4% | 94.6% |
| Mar-25 | 14 | 4.4 | 16 | 506 | 495 | −2 | −1.2% | 6.3% | 6.2% | 93.8% |
| **Grand total** | **845** | **6.1** | **218** | **65** | **495** | **627** | **684.6%** | **84.0%** | **8.2%** | ■ |
| **Campaign total** | **751** | **11.5** | **104** | **44** | **628** | **647** | **1270.5%** | **27.4%** | **7.8%** | ■ |

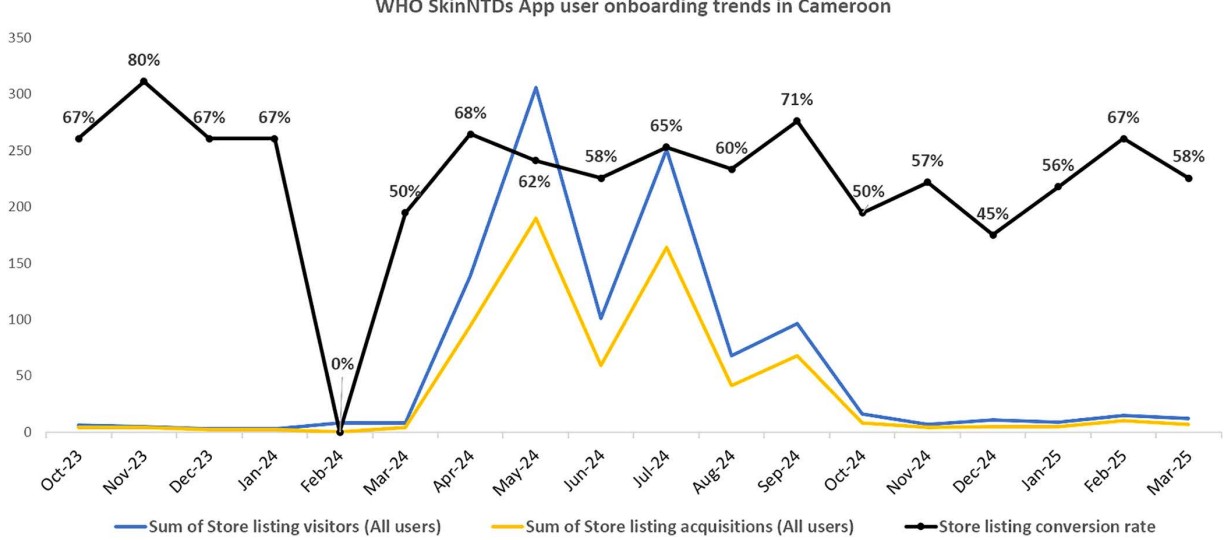

**Fig 1.  App user onboarding trends.**

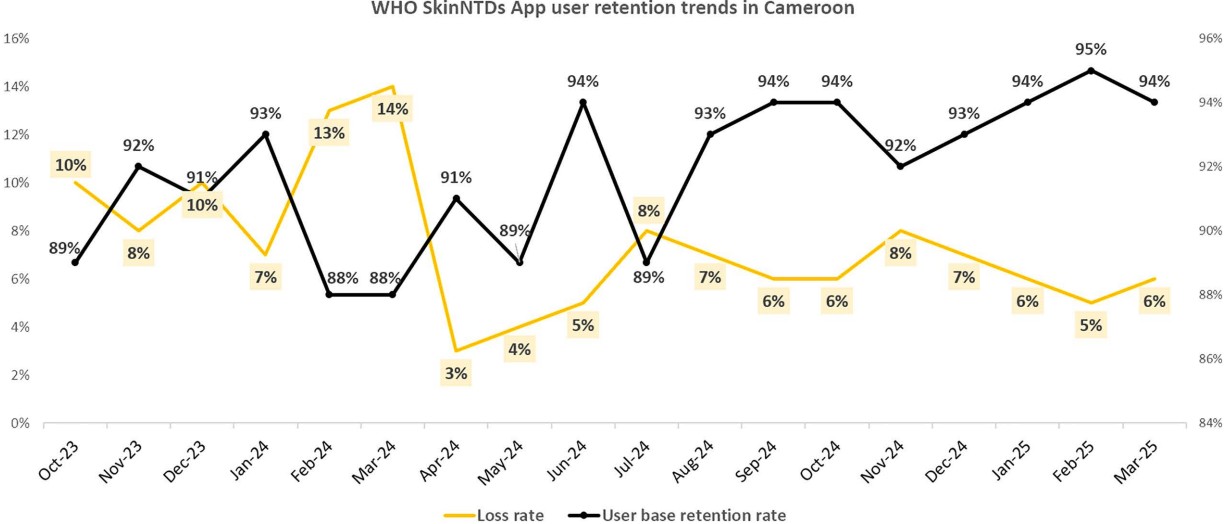

**Fig 2. App user retention trends.**

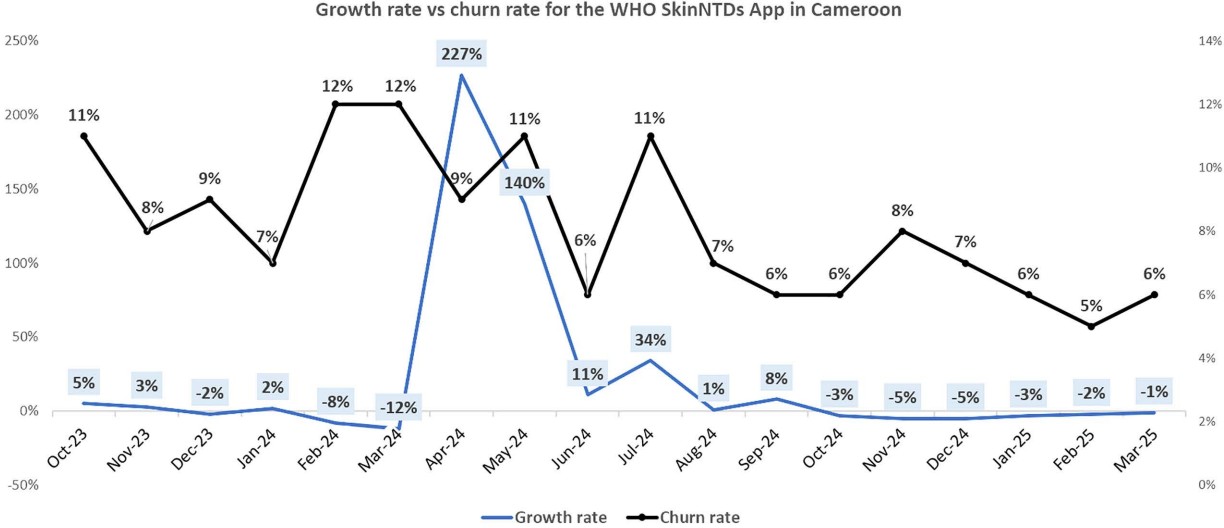

**Fig 3. App user base growth trends.**

p = 0.003, η² = 0.532), with post-campaign (p = 0.007, d = −2.093, 95% CI [−3.57, −0.62]) and campaign (p = 0.008, d = −2.035, 95% CI [−3.50, −0.57]) periods showing markedly lower churn compared to pre-campaign. Similarly, user base retention rates (F(2,15) = 8.53, p = 0.003, η² = 0.532) were significantly higher in the post-campaign (p = 0.007, d = 2.093, 95% CI [0.62, 3.57]) and campaign (p = 0.008, d = 2.035, 95% CI [0.57, 3.50]) periods relative to pre-campaign. A sensitivity analysis applying a conservative Bonferroni correction across the six outcome measures (new α = 0.05/6 = 0.0083) would not alter the interpretation of the significant period effects for growth rate (p = 0.007), churn rate (p = 0.003), and user retention (p = 0.003), as their p-values remain below the adjusted threshold. The effect for loss rate (p = 0.011) did not survive this conservative correction, while the non-significant results for conversion metrics were unchanged. Details of these statistical analyses are provided in supporting information file S5 Appendix.

### Results from interrupted time series analyses

*Impact of the SkinNTDs app dissemination campaign in Cameroon:*. The ARIMA(2,0,0) ITS model for Cameroon showed no significant pre-campaign slope (β = −0.000, p = 0.331) and a positive but non-significant immediate level change at campaign start (β = 0.096, p = 0.068), followed by a significant slope increase during the campaign (β = 0.013, p < 0.001). The post-campaign period showed a non-significant immediate level change (β = 0.130, p = 0.074) and a significant negative slope change (β = −0.016, p < 0.001). Model fit was excellent (pseudo-$R^2$ = 0.997, RMSE = 0.052 on log scale), and diagnostics indicated no residual autocorrelation or heteroscedasticity. These results suggest the campaign produced a sustained acceleration in adoption during its implementation, which reversed after its conclusion.

### Cross-country comparison to track campaign's effect on Install base

The GAMM (AIC = 1512.738, REML score = 833.460) explained 88.7% of the variance in standardized Install base trends across countries (adjusted $R^2$ = 0.885). In Cameroon, the post-intervention period was associated with a non-significant immediate level change (β = −0.130, SE = 0.086, p = 0.133) but a significant nonlinear slope change (edf = 3.921, F = 206.680, p < 0.001), indicating altered adoption trajectories after campaign initiation. Control countries (Ethiopia, Ghana, Nigeria) all exhibited significant nonlinear temporal trends for Install base (all p < 0.001) but no significant post-intervention slope changes (p = 0.179). There were no structural breaks detected in Cameroon before the campaign. These results suggest that while the campaign did not produce an immediate surge in adoption, it significantly accelerated growth relative to controls, consistent with a delayed or cumulative intervention effect. Fig 4 shows Install base trends in Cameroon, Ethiopia, Ghana, and Nigeria across study period.

### Cross-country comparison to track campaign's effect on Daily active devices

The Poisson GLMM (AIC = 9613.3, logLik = −4792.7) indicated a significant immediate increase in daily active devices (DADs) in Cameroon following the campaign (β = 3.133, SE = 0.111, p < 0.001), with a marginally non-significant negative post-intervention slope change (β = −0.001, SE = 0.001, p = 0.063). Relative to Cameroon, baseline DAD levels were

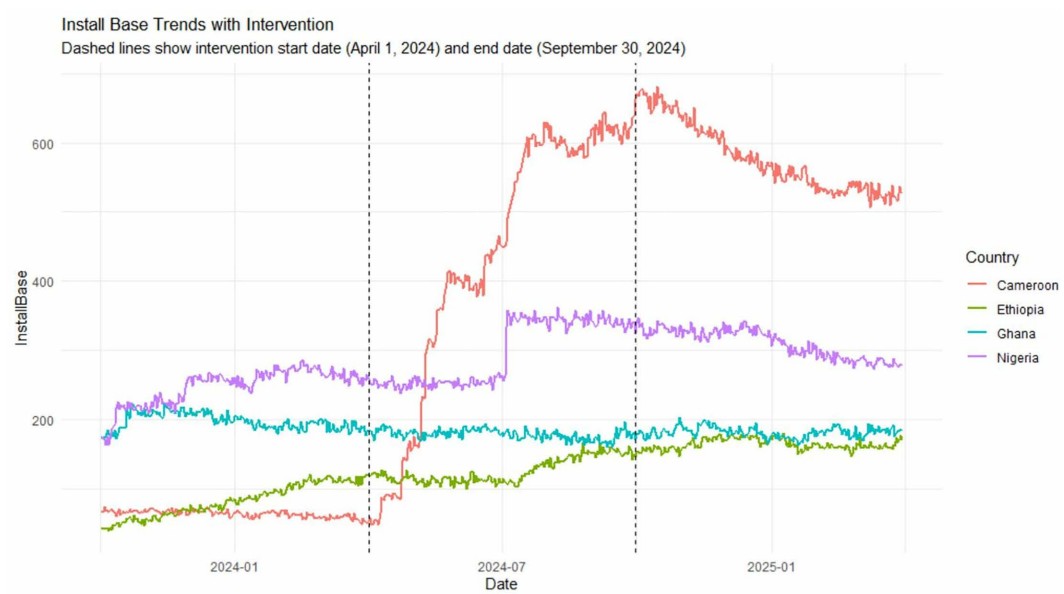

**Fig 4. Install base trends in Cameroon vs Ethiopia, Ghana and Kenya.**

higher in Ethiopia, Ghana, and Nigeria (all $p < 0.001$), but each showed significantly smaller immediate level changes after the intervention date (all $p < 0.001$). Post-intervention slopes were positive for Ethiopia ($p < 0.001$) and marginally for Ghana ($p = 0.080$), and non-significant for Nigeria ($p = 0.295$). These findings support a strong short-term campaign effect on active use in Cameroon, which was not mirrored in controls and showed signs of decline without sustained promotional activities. Detailed results of these ITS analyses are available in supporting information file S3 Appendix.

## Maximizing engagement

We attempted to maintain app engagement by activating the WhatsApp group (55 members as of September 30, 2024). Initiated in 1 May 2024, this activation consisted of sharing messages each or every other day intended to stimulate continuous use of the app by group members. For instance, we prepared and shared short videos presenting cases of app use. One of these went through the diagnostic steps to distinguish between Kaposi sarcoma and Herpes Zona. We also shared pictures of skin lesions obtained from the app as tests for participants and encouraged them to share their own images, from suspect cases seen in the field, through the app. This required participants to navigate within the app, and thus contribute to continuous learning about both sNTDs and app use.

We also encouraged group participants to interact through quizzes. Questions included the following: "*Can the WHO SkinNTDs app replace face-to-face training on the diagnosis of skin diseases and skin NTDs?*", or "*Among the steps proposed for using the WHO SkinNTDs application, which do you consider the most important for the detection of suspected cases of skin NTDs?*". WhatsApp group members were also encouraged to share resources and the app download link with other colleagues.

A timeframe for sharing messages in the WhatsApp group was created to align with recommendations from previous studies on when sending notifications through mHealth apps is most effective [43,44]. In general, this was after working hours, as it was assumed that FHWs would be more likely to use their phone for leisure pursuits after work.

## Discussion

### Discussion of principal results

The dissemination activities we implemented covered various marketing concepts and led to a cumulative number of 751 app installs. The 6-month campaign created a 3,538% increase in impressions (pre-campaign average: 7 vs campaign average: 151) and 871% increase in installed audience (pre-campaign average: 48 vs campaign average: 466). Conversion rates were consistently high (58–70%) during active campaign months, reflecting compelling store listings and targeted messaging. However, during the post-campaign phase, metrics reverted to near-baseline levels, emphasizing the need for ongoing engagement strategies. Furthermore, statistical analyses showed that while the campaign demonstrated efficacy in driving device adoption and reducing attrition, it did not influence store-specific conversion metrics.

Across descriptive metrics and complementary ITS analyses, the dissemination campaign in Cameroon was consistently associated with marked gains in SkinNTDs app adoption and use, although the magnitude and persistence of effects varied. Campaign months showed sharp increases in installations, net growth, and retention, with low churn and loss rates compared to pre- and post-campaign periods.

The ARIMA model indicated a significant acceleration in adoption during the campaign that reversed after its conclusion. The GAMM, incorporating control countries, identified a significant post-intervention slope change unique to Cameroon, consistent with a delayed or cumulative effect rather than an immediate surge. The Poisson GLMM showed a large immediate rise in daily active devices in Cameroon, absent in controls, but with indications of gradual decline thereafter.

Taken together, these findings suggest that the campaign was effective in stimulating adoption and engagement in the short term, while sustaining these gains may require continued promotional efforts.

We acknowledge secular trends (e.g., WHO advocacy) could influence results. For example, WHO NTDs team disseminated the existence of the app through its website and sending emails to our international networks, over 1000 people

in their data base. The team members also talked about the app during the meetings they organized or attended. Future studies should incorporate non-linear terms and run randomized experiments for stronger causal inference.

### Discussion on the implementation of strategies within the marketing plan

Our marketing plan featured a multi-channel approach utilizing WhatsApp, videos, in-person training, and print materials. Most of these strategies were successfully implemented: WhatsApp and videos effectively raised awareness, resulting in 64% of Play Store visitors downloading the app, and in-person training boosted engagement, with health workers being more likely to use the app when introduced to it by district officers.

We did not use print materials such as posters, flyers, and newspaper advertisements for a number of financial and logistical reasons: firstly, a one-eighth page advertisement on the back cover of a newspaper would have cost over US $150, with no guarantee that the information would reach FHWs in the most remote areas. Secondly, printed materials would have required a plan for distribution by hand. Thirdly, the message on these materials would be confusing in the event we updated our approach in response to feedback. Ultimately, we also abandoned the use of QR codes linking to the app store page. A quick informal survey among the target audience revealed that QR codes were not commonly accessed.

Concerning the objectives initially planned, we succeeded to reach 100% of health districts in both North and Far North regions, achieved a 64.1% conversion rate (vs 60% planned), and obtained 93.2% user base retention rate at 3-months post-campaign. We also achieved 12 sessions + 18 facility visits compared to a minimum of 20 in-person sessions initially planned.

### Comparison with other mHealth initiatives

Three other mHealth initiatives – the e-MCH Handbook (utilizing posters and health center staff training in Jordan) [45], the CycleBeads app (reliant on social media ads in Ghana, Egypt and other low-income countries) [46], and the Guaral app (community-driven, human-centered design with training for community health workers and nurses in Colombia) [47,48] – provide distinct and useful information about dissemination in low-resource settings. For the e-MCH Handbook, a smartphone-based maternal and child health tool for refugees, posters, flyers, and a training-of-trainers program were used to promote adoption. However, neither the dissemination methods nor staff training were sufficient to prompt significant uptake. This underscores a key challenge in mHealth implementation: passive strategies (e.g., printed materials) and one-time training often fail to sustain engagement without deeper systemic integration.

Unlike the CycleBeads app campaign, which relied on culturally tailored Facebook ads, our approach prioritized the ability of offline functionality to counteract the gaps in Cameroon's digital infrastructure [15]. Similarly, while the Guaral app was only promoted through in-person training sessions, we deployed a hybrid approach, which achieved broader reach (628 active users) but resulted in comparable challenges to long-term engagement. In our study this was due to post-training uninstalls; in the Guaral research this was due to a need for refresher training to maintain user engagement. Both studies highlight a critical element for successful campaigns in low-resource settings: context-specific adaptations.

While we segmented our target audience (e.g., frontline health workers vs. policymakers), we did not fully adopt the "*Be He@lthy, Be Mobile*" toolkit recommendation to create detailed user personas [49]. Developing such personas as "Community nurse in rural Cameroon" may have fostered greater empathy and more tailored messaging, potentially resulting in improved retention. Additionally, message relevance and implementation buy-in may have been strengthened by involving stakeholders in collaborative workshops to refine the marketing plan.

### Marketing concepts applied in the WHO SkinNTDs App dissemination plan vs. systematic review recommendations

We analyzed the various marketing concepts applied in this campaign compared to those reported in the recent systematic review of marketing strategies for mHealth applications [26]. While this pilot successfully utilized multi-channel

dissemination (WhatsApp, in-person training) and A/B testing, it lacked the remarketing and predictive strategies described in the systematic review. For instance, remarketing would have required to identify every individual who has come into contact with the app but later uninstalled it to bring them back. Personalized messaging increased short-term adoption, but a need for sustained engagement was highlighted by post-campaign attrition. Future implementations should integrate automated tools (e.g., push notifications) and embed the app within routine health activities to improve retention, as detailed in the framework for effective mHealth scaling in resource-limited settings by Labrique et al. [50]. The comparison is summarized in Table 4 below.

## Limitations

Our data on downloads, installs, and retention were collected exclusively from Google Play Console, potentially underestimating actual usage as some users may have downloaded the app from the Apple Store or got it via Xender app from their colleagues. Google Play Console metrics—such as Install Events, Install Base, DAU/MAU, Device Acquisition, Uninstall Events, and User Acquisition—capture Android installs regardless of distribution channel, including sideloaded installations via Xender app. However, none of these metrics track iOS users, as installations on Apple devices fall completely outside the Google Play ecosystem. Moreover, metrics tied to the Play Store listing (Store Listing Visitors, Store Listing Acquisitions, Device Store Listing Impressions) cannot capture sideload installations or iOS installs.

An a priori power calculation could not be conducted due to the absence of prior estimates for expected effect sizes and variability in these specific Google Play Console metrics. Also, the lack of a statistical correction across the six-outcome metrics increases the family-wise risk of Type I errors. However, we believe this risk is mitigated by the consistency of the results (e.g., the complementary changes observed in growth, churn, and retention rates), their very large effect sizes (e.g., Cohen's $d > |1.5|$ for key metrics), and their temporal alignment with the intervention period. These factors strengthen our confidence in the validity of the core findings despite this methodological limitation.

Additionally, we did not capture subnational data (e.g., regional or district levels), which would have enabled a more granular analysis of app adoption. Key engagement metrics (daily active users, monthly active users) were not available from our Google Play Console interface and we did not explore reasons for disengagement from the app. These data would have helped provide a clearer picture of engagement dynamics.

We targeted FHWs but added third-year nursing students as a proxy subpopulation, as trainings occurred in both clinical (health facilities/program meetings) and academic (nursing schools) settings. A companion study (under review) found no significant difference in the app quality mean scores between professionals and students, supporting the validity of this proxy approach.

Table 4. Comparison of marketing strategies: SkinNTDs App implementation vs. evidence-based practices.

| Marketing concept | Applied? | Comments |
| --- | --- | --- |
| Multi-Channel Approach | Yes | WhatsApp campaigns, face-to-face training, video presentations, emails. |
| Mobile App Attribution | Yes | Downloads, installs, retention rates, and uninstalls tracked via Google Play Console. |
| A/B Testing | Yes | Different versions of promotional videos tested; adjusted based on user feedback. |
| Content Marketing | Yes | Educational videos, WhatsApp group discussions (case studies, quizzes). |
| Personalization | Yes | Tailored messages to FHWs (e.g., offline functionality, WHO credibility). |
| Behavioral Marketing | Yes | Segmented WhatsApp users; encouraged app navigation through interactive content. |
| Loyalty Marketing | Partially | Created a WhatsApp group for engagement but lacked structured rewards/updates. |
| Incentivization | Partially | Offered technical support but no monetary incentives for downloads. |
| Remarketing/Retargeting | No | No strategies to re-engage users who uninstalled the app. |
| Programmatic Marketing | No | No automated ad bidding or real-time ad adjustments. |
| Predictive Marketing | No | No data-driven forecasts to optimize engagement timing or user retention. |

Observer bias (Hawthorne effect) [51], may have occurred due to active engagement via WhatsApp and in-person sessions, potentially inflating short-term app usage. While post-campaign attrition suggests this effect was temporary, we cannot rule out behavioral modifications (e.g., increased self-reporting). Future studies should minimize observer presence and verify user roles through in-app registration.

Beyond the main introductory video, we did not apply a systematic refinement process to other use-case videos, dissemination messages, or PowerPoint presentations. In addition, we cannot confirm whether Video B had a greater impact on app adoption and campaign metrics than Video A. Future implementations should adopt a standardized A/B testing approach for all dissemination materials.

While we minimized bias in targeting FHWs during dissemination activities, we cannot exclude the possibility that some users who visited the app page or downloaded were not FHWs. For instance, other individuals could have been informed about the app through posts on WhatsApp or LinkedIn. Furthermore, while we tracked overall downloads and uninstalls, we could not attribute these outcomes to specific dissemination channels or assess user motivations. Qualitative insights on app usability are reported elsewhere. These limitations highlight the need for improved data collection methods in future studies.

The marketing strategy was implemented independently of routine NTD activities, such as mass drug administration (MDA) campaigns, which may have impacted health worker receptivity and app adoption. To maximize engagement and adoption, future efforts should align app dissemination with routine NTD campaigns, such as the annual community-directed treatment with ivermectin and deworming campaigns. These campaigns, which span four to six months and involve extensive FHW participation, including community health workers, provide an ideal opportunity for app promotion.

We agree that the 6-month follow-up period limits assessment of long-term sustainability. While this timeframe may suffice to evaluate adoption during a time-bound mass drug administration (MDA) campaign for NTDs—where short-term engagement is critical—it is insufficient to gauge sustained use beyond the intervention phase. Integration into national programs (e.g., recurring MDA campaigns) could improve long-term viability. Future studies should track retention over extended periods (≥12 months) to assess institutionalization.

Attendance sheets for workshops were maintained. However, information on how the fidelity (i.e., whether activities were delivered as planned) or actual exposure of participants to each channel was not systematically monitored. Thus, exposure data were not linked to individual app-usage metrics. This should be tracked into future studies.

We acknowledge the importance of cost-effectiveness data for policy decisions. A formal cost-effectiveness analysis was precluded by the app's early-stage evaluation (WHO stages 1–2 of digital intervention maturity) [52]. As this was a proof-of-concept dissemination study, we prioritized adoption metrics over economic evaluation. Future scale-up studies should incorporate metrics such as cost-per-install, or cost-per-case detected, particularly when transitioning to WHO stages 5–6 (Scale-up/Integration/sustainability).

## Knowledge gained

1. Video and in-person presentations are effective for conveying information and encouraging people to search for and download the app.

2. When introduced by a district management officer, health area chiefs are more likely to download the app.

3. Sharing direct download links on WhatsApp is effective but could require that the sending contact is registered on receivers' phone if sent individually; otherwise, the link should be shared in a WhatsApp group.

4. WhatsApp groups can contribute to sustaining engagement with the app and continuous surveillance.

## Policy implications and recommendations

### 1. Scaling up the app

Scaling up dissemination of the app would require a well-designed and context-adapted strategy, implemented in collaboration with major national NTD stakeholders. The near 1:1 user-to-device ratio in this study suggests each FHW is using the app on their own device, which eases the capacity-building process and long-term adoption at the individual level. Integrating NTD programs with other health system campaigns would also increase app adoption.

### 2. Integration into national health systems

By bridging the localized approach used with the Guaral app [47] and the mass reach of the CycleBeads campaign [46], our WhatsApp + on-site training model provides a balanced strategy for dissemination within LMICs. In our field experiment, we noted that chiefs of health areas were more inclined to download the app when introduced to it by a district management officer. We recommend that dissemination of the WHO SkinNTDs app is integrated within other health system activities, following the health system hierarchy. In Cameroon for instance, regional NTD program focal points are members of the regional supervisory pool for activities carried out in the region. Hence, they could easily negotiate a joint briefing with other health programs. At district level, management officers are permitted to arrange communications about any topic they see fit. Finally, the purpose of the app must match the objectives of any national digital health strategic plan, as app implementation could be vetoed by high level stakeholders if it does not align with national strategy [15].

### 3. Recommendations for developing a marketing plan

A marketing plan for an app such as WHO SkinNTDs should be managed like any organizational workplan. While it could initially be drafted by an individual or small group, this draft should be presented and discussed within the organization or with key stakeholders to boost contextualization, drive key stakeholder appropriation, and smooth adoption.

### 4. Recommendations for implementing a marketing plan

In this study it was hard to keep people engaged with the WhatsApp group, especially when content was not regularly shared and there was a lack of engaging conversations. If this continued for some time, people would start asking what was happening and increasingly leave the group. There were also instances of "quiet quitting" as people no longer interacted, with some deactivating or uninstalling the app. Thus, we make two recommendations:

(i) Maximize WhatsApp group communications during a specific timeframe. This could be during MDA campaigns for NTDs or during NTD case-finding activities in the field. Our experience suggests a timeframe period of three to six months.

(ii) Ideally, only enroll health personnels in the same region into the group.

## Contributions of this study

The main contributions from this study relate to methodology, the implementation of marketing plans for mHealth apps in general, and for not-for-profit mHealth apps in particular. Although this study involved an mHealth app targeted at health professionals, our approach could be adapted for mHealth apps aimed at patients and the general population. With an increasing number of NTDs mHealth apps, especially using artificial intelligence to identify sNTDs, we believe mHealth app implementers will benefit from the approach to dissemination taken in this study.

During our preliminary research, we noted a scarcity of advice among the academic literature, with the majority of available resources originating from and oriented towards the business world, and referring to mobile apps in general. In this paper, we have attempted to provide detailed guidance for designing, implementing, evaluating and reporting a marketing plan for mHealth apps.

As part of our efforts to raise awareness about the WHO SkinNTDs app in Cameroon and its potential contribution to skin diseases detection, preliminary observations from the field indicate that during the six-month campaign period, FHWs started to share pictures of suspected cases of diverse skin diseases in the WhatsApp group and request confirmation from peers and supervisors.

### Future directions

Tracking changes in practice among FHWs who adopted the app was beyond the objectives of this study. Further research is required to explore long-term user engagement and the degree to which the app contributes to clinical effectiveness and sNTDs surveillance.

Building on this study, future research could also include the development of standardized guidelines for disseminating mHealth apps that aid in the diagnosis of skin-related diseases. Such work could be adapted from existing guidelines, such as the Texla telehealth resource center toolkit [27]. We foresee the resulting guidelines mirroring the structure of the "*Be He@lthy, Be Mobile*" handbooks, which offer content and operational guidance for planning and implementing national mHealth programs [53]. These handbooks cover five main areas: operations management, content development and adaptation, promotion and recruitment, technology specifications, and monitoring and evaluation.

### Conclusion

This implementation of a multi-channel marketing strategy for the WHO SkinNTDs app in Cameroon has demonstrated that social media platforms like WhatsApp are effective for raising awareness and driving downloads. However, on-site in-person training could have greater impact on engagement and adoption among FHWs, especially in areas with significant barriers to digital infrastructure.

While the app achieved 628 active users and a 64% conversion rate, maintaining user retention and preventing uninstalls require sustained engagement strategies. Future efforts should focus on integrating the app into national health systems and leveraging existing sNTD programs to ensure widespread adoption. Additionally, further research is needed to evaluate the long-term contribution of the app to sNTD surveillance and its impact on improving health outcomes in endemic regions.

With an emphasis on tailored marketing strategies and stakeholder collaboration, this study provides a roadmap for the successful dissemination of mHealth tools in low-resource settings. Addressing barriers to adoption and fostering continuous engagement will enable the WHO SkinNTDs app to significantly enhance the capacity of FHWs in the management of sNTDs.

### Supporting information

**S1 Appendix. Descriptive of video content and refinement process.**
(DOCX)

**S2 Appendix. Definitions of key Google Play Console Analytics metrics.**
(DOCX)

**S3 Appendix. Interrupted Time Series Analyses.**
(DOCX)

**S4 Appendix. A draft of marketing plan for WHO SkinNTDs. app.**
(DOCX)

**S5 Appendix. Detailed results of statistical tests.**
(DOCX)

**S1 Checklist. Inclusivity in global research checklist.**
(DOCX)

**S1 Dataset. Data derived from Google Play Console for the WHO SkinNTDs. app.**
(XLSX)

## Acknowledgments

We are immensely grateful to the healthcare professionals, including those at regional and district level, who contributed to this work.

## Author contributions

**Conceptualization:** Henri Claude Moungui, Jose Antonio Ruiz-Postigo, Carme Carrion.

**Data curation:** Henri Claude Moungui.

**Formal analysis:** Henri Claude Moungui.

**Funding acquisition:** Henri Claude Moungui, Carme Carrion.

**Investigation:** Henri Claude Moungui, Paul Tonkoung Iyawa.

**Methodology:** Henri Claude Moungui, Hugues Nana-Djeunga, Carme Carrion.

**Project administration:** Henri Claude Moungui.

**Resources:** Henri Claude Moungui.

**Supervision:** Hugues Nana-Djeunga, Jose Antonio Ruiz-Postigo, Carme Carrion.

**Validation:** Carme Carrion.

**Writing – original draft:** Henri Claude Moungui, Paul Tonkoung Iyawa.

**Writing – review & editing:** Henri Claude Moungui, Paul Tonkoung Iyawa, Hugues Nana-Djeunga, Jose Antonio Ruiz-Postigo, Carme Carrion.

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
