## [Decision Letter · Decision Letter 0]

23 Jul 2025

PONE-D-25-31736Implementation of a marketing plan for the dissemination of the WHO SkinNTDs app in CameroonPLOS ONE

Dear Dr. Moungui, Thank you for submitting your manuscript to PLOS ONE. After careful consideration, we feel that it has merit but does not fully meet PLOS ONE’s publication criteria as it currently stands. Therefore, we invite you to submit a revised version of the manuscript that addresses the points raised during the review process.

**Comments**

**Reviewer 1 **

This is a well-written and timely manuscript addressing an important issue: the dissemination of a digital health tool to improve sNTD diagnosis in a low-resource setting. The implementation of a marketing strategy tailored to frontline health workers (FHWs) in Cameroon provides a valuable contribution to the literature on mHealth interventions in LMIC contexts.

However, the paper in its current form presents several methodological limitations and reporting gaps that limit the validity and generalizability of the conclusions.

1. Short Follow-Up Period

The follow-up period post-intervention is limited to six months. This is insufficient to assess the long-term sustainability of the intervention, especially since most gains appear concentrated in the campaign phase. A longer follow-up or at least a more detailed discussion of sustainability challenges is warranted.

2. Population Selection Bias and Observer Effect

The manuscript acknowledges the possibility that some users may not be FHWs, but this issue is underexplored. Additionally, since many participants were actively engaged through WhatsApp and in-person presentations, there is a high risk of observer effect (Hawthorne effect), where participants modify behavior due to awareness of being observed, which is not discussed sufficiently.

3. Lack of a Control Group

Without a control group (e.g., a comparable region where no dissemination activities were conducted), it is difficult to attribute observed increases in app downloads or retention to the intervention rather than external factors or secular trends. A stronger justification of the quasi-experimental design, or acknowledgment of this limitation, is necessary.

4. Exclusive Reliance on Google Play Data

All quantitative app engagement metrics are derived exclusively from Google Play Console, ignoring iOS users and informal app-sharing mechanisms common in Low- and Middle-Income Countries settings (e.g., Xender). This likely results in underreporting of actual app adoption. The authors should clarify the proportion of Android vs. iOS devices in the target population and better justify the reliance on Android data alone.

5. No Measure of Clinical Impact

The study focuses entirely on process outcomes (downloads, installs, retention) without assessing any impact on clinical practice or health outcomes (e.g., improved diagnosis or reporting of sNTDs). The absence of even a basic qualitative assessment of changes in practice is a significant limitation that must be more thoroughly acknowledged.

6. No Cost-Effectiveness Analysis

There is no data on cost per download, cost per retained user, or resource expenditure relative to outcomes. Given the relevance of cost-effectiveness for decision-making in LMIC health systems, this represents a missed opportunity. At minimum, the authors should discuss this limitation more openly. While the app is freely available, the implementation of the marketing strategy incurred costs (staff salaries, transportation, in-person events, communication materials). The paper does not account for these costs relative to key outcomes (e.g., cost per additional active user, cost per retained user). Without this, policymakers in similar settings cannot assess the affordability or scalability of the approach. I recommend either a basic cost analysis or a transparent acknowledgment of this omission in the limitations.

7. Use of Basic Statistical Tools

The choice of Microsoft Excel and Jamovi for statistical analysis raises concerns about analytical robustness. While acceptable for basic ANOVA and non-parametric tests, the use of more sophisticated statistical packages (e.g., R or STATA) could enable richer analyses such as time-series modeling or interrupted time series designs, which seem more appropriate for before-after quasi-experimental data.

8. Lack of Clarity on p-Value Correction

The manuscript does not sufficiently describe whether p-value corrections for multiple testing were applied, especially given multiple outcome metrics. While the use of Dwass-Steel-Critchlow-Fligner tests is appropriate for pairwise comparisons, there is no clarity on overall correction for Type I error across multiple comparisons, which is critical in studies with numerous endpoints.

9. Inadequate Insight on WhatsApp Group Dynamics

The paper rightly identifies WhatsApp as a key engagement tool, yet offers limited analysis of group activity (e.g., participation rates, message frequency, engagement levels over time). Some quantitative or qualitative reporting on WhatsApp engagement would substantially strengthen the insights about sustained app use.

Additional Observations:

Reference [3] appears in the reference list but is not cited in the main text. This should be corrected.

Some sentences would benefit from minor grammatical editing for conciseness and clarity.

Statistical significance is sometimes highlighted without consistent emphasis on effect sizes or practical significance, which would be more informative to readers.

**Reviewer 2**

The description of the marketing plan outlines various dissemination channels (WhatsApp, printed posters, face‐to‐face training, emails) but does not anchor these activities in a recognized health behaviour or communication theory (e.g., Health Belief Model, Diffusion of Innovations). Explicitly identify and describe a theoretical framework that guided message development and channel selection. This will strengthen the rationale for why certain campaign components were expected to influence FHW knowledge, attitudes, or practices.

¾ Although a “marketing plan” is mentioned, there is no clear statement of SMART (Specific, Measurable, Achievable, Relevant, Time‐bound) objectives (e.g., “Increase app downloads among FHWs in Region X by 25% within three months”).Include a table listing precise campaign objectives (e.g., target reach, frequency of message exposure, engagement metrics) and corresponding process and outcome indicators (e.g., number of posters displayed, number of WhatsApp broadcasts sent, click‐through rates).

¾ The methods lack any mention of formative assessment (e.g., focus groups, key‐informant interviews) or pretesting of messages/materials with the target audience before full rollout. Describe any formative work undertaken to tailor content (language, visuals) to the local context and document how A/B testing informed final message versions. If no formative research was done, discuss this as a limitation.

¾ There is no information on how the fidelity (i.e., whether activities were delivered as planned) or actual exposure of participants to each channel was monitored. Detail procedures for tracking campaign delivery (e.g., logs of WhatsApp broadcast times, attendance sheets for workshops, distribution records for posters) and how exposure data were linked to individual app‐usage metrics.

¾ The before–after design without a contemporaneous control group is vulnerable to secular trends (e.g., global publicity, other local NTD initiatives) that might influence app uptake.Acknowledge and discuss potential confounders (e.g., simultaneous WHO or NGO activities), and consider whether any comparison districts without targeted marketing could have been used as controls.

¾ The methods state that “videos and presentations” were used, but offer no specifics on content length, format, language, or whether materials were adapted for literacy levels. Provide sample outlines or screenshots of videos/slides (e.g., durations, main messages), and describe how materials were localized (translations, pictograms) for FHW audiences.

¾ While the use of ANOVA and non‐parametric tests is described, there is no mention of how missing data (e.g., uninstalls, silent non-users) were handled or whether any adjustments were made for multiple comparisons across three periods.Specify data-cleaning rules, handling of outliers, and correction methods (e.g., Bonferroni) if multiple pairwise contrasts were conducted. Consider presenting an a priori power calculation for detecting meaningful changes in download or retention rates.

¾ The six‐month pre-, during-, and post-campaign windows are clear, but dates in the text (e.g., “October 2024–March 2025”) should be formatted consistently (e.g., “1 October 2024 to 31 March 2025”).

¾ Provide specifics on how A/B tests were designed (randomization process, sample sizes per variant, success criteria), and indicate which version emerged as optimal.

¾ Explain how frontline health workers were identified and invited to participate in the campaign—was it purposive, convenience, or exhaustive sampling within target districts?

Please submit your revised manuscript by Sep 06 2025 11:59PM. If you will need more time than this to complete your revisions, please reply to this message or contact the journal office at plosone@plos.org. Please include the following items when submitting your revised manuscript:

We look forward to receiving your revised manuscript.

Kind regards,

Fatemeh Zarei, PhD

Academic Editor

PLOS ONE

2. In your Methods section, please ensure that you have included a statement specifying whether the collection and analysis method complied with the terms and conditions for the source of the data.

3. Please include a complete copy of PLOS’ questionnaire on inclusivity in global research in your revised manuscript. Our policy for research in this area aims to improve transparency in the reporting of research performed outside of researchers’ own country or community. The policy applies to researchers who have travelled to a different country to conduct research, research with Indigenous populations or their lands, and research on cultural artefacts. The questionnaire can also be requested at the journal’s discretion for any other submissions, even if these conditions are not met.  Please find more information on the policy and a link to download a blank copy of the questionnaire here: https://journals.plos.org/plosone/s/best-practices-in-research-reporting. Please upload a completed version of your questionnaire as Supporting Information when you resubmit your manuscript.

4. In the ethics statement in the Methods, you have specified that verbal consent was obtained. Please provide additional details regarding how this consent was documented and witnessed, and state whether this was approved by the IRB.

6. We note that Figures 2-3 in your submission contain [map/satellite] images which may be copyrighted. All PLOS content is published under the Creative Commons Attribution License (CC BY 4.0), which means that the manuscript, images, and Supporting Information files will be freely available online, and any third party is permitted to access, download, copy, distribute, and use these materials in any way, even commercially, with proper attribution. For these reasons, we cannot publish previously copyrighted maps or satellite images created using proprietary data, such as Google software (Google Maps, Street View, and Earth). For more information, see our copyright guidelines: http://journals.plos.org/plosone/s/licenses-and-copyright.

1. You may seek permission from the original copyright holder of Figures 2-3 to publish the content specifically under the CC BY 4.0 license. 

7. Please remove all personal information, ensure that the data shared are in accordance with participant consent, and re-upload a fully anonymized data set.

Comments

Reviewer 1

This is a well-written and timely manuscript addressing an important issue: the dissemination of a digital health tool to improve sNTD diagnosis in a low-resource setting. The implementation of a marketing strategy tailored to frontline health workers (FHWs) in Cameroon provides a valuable contribution to the literature on mHealth interventions in LMIC contexts.

However, the paper in its current form presents several methodological limitations and reporting gaps that limit the validity and generalizability of the conclusions.

1. Short Follow-Up Period

The follow-up period post-intervention is limited to six months. This is insufficient to assess the long-term sustainability of the intervention, especially since most gains appear concentrated in the campaign phase. A longer follow-up or at least a more detailed discussion of sustainability challenges is warranted.

2. Population Selection Bias and Observer Effect

The manuscript acknowledges the possibility that some users may not be FHWs, but this issue is underexplored. Additionally, since many participants were actively engaged through WhatsApp and in-person presentations, there is a high risk of observer effect (Hawthorne effect), where participants modify behavior due to awareness of being observed, which is not discussed sufficiently.

3. Lack of a Control Group

Without a control group (e.g., a comparable region where no dissemination activities were conducted), it is difficult to attribute observed increases in app downloads or retention to the intervention rather than external factors or secular trends. A stronger justification of the quasi-experimental design, or acknowledgment of this limitation, is necessary.

4. Exclusive Reliance on Google Play Data

All quantitative app engagement metrics are derived exclusively from Google Play Console, ignoring iOS users and informal app-sharing mechanisms common in Low- and Middle-Income Countries settings (e.g., Xender). This likely results in underreporting of actual app adoption. The authors should clarify the proportion of Android vs. iOS devices in the target population and better justify the reliance on Android data alone.

5. No Measure of Clinical Impact

The study focuses entirely on process outcomes (downloads, installs, retention) without assessing any impact on clinical practice or health outcomes (e.g., improved diagnosis or reporting of sNTDs). The absence of even a basic qualitative assessment of changes in practice is a significant limitation that must be more thoroughly acknowledged.

6. No Cost-Effectiveness Analysis

There is no data on cost per download, cost per retained user, or resource expenditure relative to outcomes. Given the relevance of cost-effectiveness for decision-making in LMIC health systems, this represents a missed opportunity. At minimum, the authors should discuss this limitation more openly. While the app is freely available, the implementation of the marketing strategy incurred costs (staff salaries, transportation, in-person events, communication materials). The paper does not account for these costs relative to key outcomes (e.g., cost per additional active user, cost per retained user). Without this, policymakers in similar settings cannot assess the affordability or scalability of the approach. I recommend either a basic cost analysis or a transparent acknowledgment of this omission in the limitations.

7. Use of Basic Statistical Tools

The choice of Microsoft Excel and Jamovi for statistical analysis raises concerns about analytical robustness. While acceptable for basic ANOVA and non-parametric tests, the use of more sophisticated statistical packages (e.g., R or STATA) could enable richer analyses such as time-series modeling or interrupted time series designs, which seem more appropriate for before-after quasi-experimental data.

8. Lack of Clarity on p-Value Correction

The manuscript does not sufficiently describe whether p-value corrections for multiple testing were applied, especially given multiple outcome metrics. While the use of Dwass-Steel-Critchlow-Fligner tests is appropriate for pairwise comparisons, there is no clarity on overall correction for Type I error across multiple comparisons, which is critical in studies with numerous endpoints.

9. Inadequate Insight on WhatsApp Group Dynamics

The paper rightly identifies WhatsApp as a key engagement tool, yet offers limited analysis of group activity (e.g., participation rates, message frequency, engagement levels over time). Some quantitative or qualitative reporting on WhatsApp engagement would substantially strengthen the insights about sustained app use.

Additional Observations:

Reference [3] appears in the reference list but is not cited in the main text. This should be corrected.

Some sentences would benefit from minor grammatical editing for conciseness and clarity.

Statistical significance is sometimes highlighted without consistent emphasis on effect sizes or practical significance, which would be more informative to readers.

Reviewer 2

The description of the marketing plan outlines various dissemination channels (WhatsApp, printed posters, face‐to‐face training, emails) but does not anchor these activities in a recognized health behaviour or communication theory (e.g., Health Belief Model, Diffusion of Innovations). Explicitly identify and describe a theoretical framework that guided message development and channel selection. This will strengthen the rationale for why certain campaign components were expected to influence FHW knowledge, attitudes, or practices.

Although a “marketing plan” is mentioned, there is no clear statement of SMART (Specific, Measurable, Achievable, Relevant, Time‐bound) objectives (e.g., “Increase app downloads among FHWs in Region X by 25% within three months”).Include a table listing precise campaign objectives (e.g., target reach, frequency of message exposure, engagement metrics) and corresponding process and outcome indicators (e.g., number of posters displayed, number of WhatsApp broadcasts sent, click‐through rates).

The methods lack any mention of formative assessment (e.g., focus groups, key‐informant interviews) or pretesting of messages/materials with the target audience before full rollout. Describe any formative work undertaken to tailor content (language, visuals) to the local context and document how A/B testing informed final message versions. If no formative research was done, discuss this as a limitation.

There is no information on how the fidelity (i.e., whether activities were delivered as planned) or actual exposure of participants to each channel was monitored. Detail procedures for tracking campaign delivery (e.g., logs of WhatsApp broadcast times, attendance sheets for workshops, distribution records for posters) and how exposure data were linked to individual app‐usage metrics.

The before–after design without a contemporaneous control group is vulnerable to secular trends (e.g., global publicity, other local NTD initiatives) that might influence app uptake.Acknowledge and discuss potential confounders (e.g., simultaneous WHO or NGO activities), and consider whether any comparison districts without targeted marketing could have been used as controls.

The methods state that “videos and presentations” were used, but offer no specifics on content length, format, language, or whether materials were adapted for literacy levels. Provide sample outlines or screenshots of videos/slides (e.g., durations, main messages), and describe how materials were localized (translations, pictograms) for FHW audiences.

While the use of ANOVA and non‐parametric tests is described, there is no mention of how missing data (e.g., uninstalls, silent non-users) were handled or whether any adjustments were made for multiple comparisons across three periods.Specify data-cleaning rules, handling of outliers, and correction methods (e.g., Bonferroni) if multiple pairwise contrasts were conducted. Consider presenting an a priori power calculation for detecting meaningful changes in download or retention rates.

The six‐month pre-, during-, and post-campaign windows are clear, but dates in the text (e.g., “October 2024–March 2025”) should be formatted consistently (e.g., “1 October 2024 to 31 March 2025”).

Provide specifics on how A/B tests were designed (randomization process, sample sizes per variant, success criteria), and indicate which version emerged as optimal.

Explain how frontline health workers were identified and invited to participate in the campaign—was it purposive, convenience, or exhaustive sampling within target districts?

Reviewers' comments:

Reviewer's Responses to Questions

**Comments to the Author**

1. Is the manuscript technically sound, and do the data support the conclusions?

Reviewer #1: Partly

Reviewer #2: Partly

2. Has the statistical analysis been performed appropriately and rigorously? 

Reviewer #1: No

Reviewer #2: I Don't Know

3. Have the authors made all data underlying the findings in their manuscript fully available?

Reviewer #1: Yes

Reviewer #2: Yes

4. Is the manuscript presented in an intelligible fashion and written in standard English?

Reviewer #1: Yes

Reviewer #2: Yes

5. Review Comments to the Author

Reviewer #1: This is a well-written and timely manuscript addressing an important issue: the dissemination of a digital health tool to improve sNTD diagnosis in a low-resource setting. The implementation of a marketing strategy tailored to frontline health workers (FHWs) in Cameroon provides a valuable contribution to the literature on mHealth interventions in LMIC contexts.

However, the paper in its current form presents several methodological limitations and reporting gaps that limit the validity and generalizability of the conclusions.

1. Short Follow-Up Period

The follow-up period post-intervention is limited to six months. This is insufficient to assess the long-term sustainability of the intervention, especially since most gains appear concentrated in the campaign phase. A longer follow-up or at least a more detailed discussion of sustainability challenges is warranted.

2. Population Selection Bias and Observer Effect

The manuscript acknowledges the possibility that some users may not be FHWs, but this issue is underexplored. Additionally, since many participants were actively engaged through WhatsApp and in-person presentations, there is a high risk of observer effect (Hawthorne effect), where participants modify behavior due to awareness of being observed, which is not discussed sufficiently.

3. Lack of a Control Group

Without a control group (e.g., a comparable region where no dissemination activities were conducted), it is difficult to attribute observed increases in app downloads or retention to the intervention rather than external factors or secular trends. A stronger justification of the quasi-experimental design, or acknowledgment of this limitation, is necessary.

4. Exclusive Reliance on Google Play Data

All quantitative app engagement metrics are derived exclusively from Google Play Console, ignoring iOS users and informal app-sharing mechanisms common in Low- and Middle-Income Countries settings (e.g., Xender). This likely results in underreporting of actual app adoption. The authors should clarify the proportion of Android vs. iOS devices in the target population and better justify the reliance on Android data alone.

5. No Measure of Clinical Impact

The study focuses entirely on process outcomes (downloads, installs, retention) without assessing any impact on clinical practice or health outcomes (e.g., improved diagnosis or reporting of sNTDs). The absence of even a basic qualitative assessment of changes in practice is a significant limitation that must be more thoroughly acknowledged.

6. No Cost-Effectiveness Analysis

There is no data on cost per download, cost per retained user, or resource expenditure relative to outcomes. Given the relevance of cost-effectiveness for decision-making in LMIC health systems, this represents a missed opportunity. At minimum, the authors should discuss this limitation more openly. While the app is freely available, the implementation of the marketing strategy incurred costs (staff salaries, transportation, in-person events, communication materials). The paper does not account for these costs relative to key outcomes (e.g., cost per additional active user, cost per retained user). Without this, policymakers in similar settings cannot assess the affordability or scalability of the approach. I recommend either a basic cost analysis or a transparent acknowledgment of this omission in the limitations.

7. Use of Basic Statistical Tools

The choice of Microsoft Excel and Jamovi for statistical analysis raises concerns about analytical robustness. While acceptable for basic ANOVA and non-parametric tests, the use of more sophisticated statistical packages (e.g., R or STATA) could enable richer analyses such as time-series modeling or interrupted time series designs, which seem more appropriate for before-after quasi-experimental data.

8. Lack of Clarity on p-Value Correction

The manuscript does not sufficiently describe whether p-value corrections for multiple testing were applied, especially given multiple outcome metrics. While the use of Dwass-Steel-Critchlow-Fligner tests is appropriate for pairwise comparisons, there is no clarity on overall correction for Type I error across multiple comparisons, which is critical in studies with numerous endpoints.

9. Inadequate Insight on WhatsApp Group Dynamics

The paper rightly identifies WhatsApp as a key engagement tool, yet offers limited analysis of group activity (e.g., participation rates, message frequency, engagement levels over time). Some quantitative or qualitative reporting on WhatsApp engagement would substantially strengthen the insights about sustained app use.

Additional Observations:

Reference [3] appears in the reference list but is not cited in the main text. This should be corrected.

Some sentences would benefit from minor grammatical editing for conciseness and clarity.

Statistical significance is sometimes highlighted without consistent emphasis on effect sizes or practical significance, which would be more informative to readers.

Reviewer #2: � The description of the marketing plan outlines various dissemination channels (WhatsApp, printed posters, face‐to‐face training, emails) but does not anchor these activities in a recognized health behaviour or communication theory (e.g., Health Belief Model, Diffusion of Innovations). Explicitly identify and describe a theoretical framework that guided message development and channel selection. This will strengthen the rationale for why certain campaign components were expected to influence FHW knowledge, attitudes, or practices.

Although a “marketing plan” is mentioned, there is no clear statement of SMART (Specific, Measurable, Achievable, Relevant, Time‐bound) objectives (e.g., “Increase app downloads among FHWs in Region X by 25% within three months”).Include a table listing precise campaign objectives (e.g., target reach, frequency of message exposure, engagement metrics) and corresponding process and outcome indicators (e.g., number of posters displayed, number of WhatsApp broadcasts sent, click‐through rates).

The methods lack any mention of formative assessment (e.g., focus groups, key‐informant interviews) or pretesting of messages/materials with the target audience before full rollout. Describe any formative work undertaken to tailor content (language, visuals) to the local context and document how A/B testing informed final message versions. If no formative research was done, discuss this as a limitation.

There is no information on how the fidelity (i.e., whether activities were delivered as planned) or actual exposure of participants to each channel was monitored. Detail procedures for tracking campaign delivery (e.g., logs of WhatsApp broadcast times, attendance sheets for workshops, distribution records for posters) and how exposure data were linked to individual app‐usage metrics.

The before–after design without a contemporaneous control group is vulnerable to secular trends (e.g., global publicity, other local NTD initiatives) that might influence app uptake.Acknowledge and discuss potential confounders (e.g., simultaneous WHO or NGO activities), and consider whether any comparison districts without targeted marketing could have been used as controls.

The methods state that “videos and presentations” were used, but offer no specifics on content length, format, language, or whether materials were adapted for literacy levels. Provide sample outlines or screenshots of videos/slides (e.g., durations, main messages), and describe how materials were localized (translations, pictograms) for FHW audiences.

While the use of ANOVA and non‐parametric tests is described, there is no mention of how missing data (e.g., uninstalls, silent non-users) were handled or whether any adjustments were made for multiple comparisons across three periods.Specify data-cleaning rules, handling of outliers, and correction methods (e.g., Bonferroni) if multiple pairwise contrasts were conducted. Consider presenting an a priori power calculation for detecting meaningful changes in download or retention rates.

The six‐month pre-, during-, and post-campaign windows are clear, but dates in the text (e.g., “October 2024–March 2025”) should be formatted consistently (e.g., “1 October 2024 to 31 March 2025”).

Provide specifics on how A/B tests were designed (randomization process, sample sizes per variant, success criteria), and indicate which version emerged as optimal.

Explain how frontline health workers were identified and invited to participate in the campaign—was it purposive, convenience, or exhaustive sampling within target districts?

6. PLOS authors have the option to publish the peer review history of their article (what does this mean?). If published, this will include your full peer review and any attached files.

Reviewer #1: No

Reviewer #2: No

---

## [Author Response · Author response to Decision Letter 1]

18 Aug 2025

Point-by-point responses to reviewer 1

1. Short Follow-Up Period : The follow-up period post-intervention is limited to six months. This is insufficient to assess the long-term sustainability of the intervention, especially since most gains appear concentrated in the campaign phase. A longer follow-up or at least a more detailed discussion of sustainability challenges is warranted.

Response:

We have acknowledged the limitation and expanded discussion: We agree that the 6-month follow-up period limits assessment of long-term sustainability. While this timeframe may suffice to evaluate adoption during a time-bound mass drug administration (MDA) campaign for NTDs—where short-term engagement is critical—it is insufficient to gauge sustained use beyond the intervention phase. Integration into national programs (e.g., recurring MDA campaigns) could improve long-term viability. Future studies should track retention over extended periods (≥12 months) to assess institutionalization.

2. Population Selection Bias and Observer Effect: The manuscript acknowledges the possibility that some users may not be FHWs, but this issue is underexplored. Additionally, since many participants were actively engaged through WhatsApp and in-person presentations, there is a high risk of observer effect (Hawthorne effect), where participants modify behavior due to awareness of being observed, which is not discussed sufficiently.

Response:

Thank you for this suggestion. We have added the following in Limitations: We targeted FHWs but included third-year nursing students as a proxy subpopulation, as trainings occurred in both clinical (health facilities/program meetings) and academic (nursing schools) settings. A companion study (under review) found no significant difference in the app quality mean scores between professionals and students, supporting the validity of this proxy approach.

Observer bias (Hawthorne effect), may have occurred due to active engagement via WhatsApp and in-person sessions, potentially inflating short-term app usage. While post-campaign attrition suggests this effect was temporary, we cannot rule out behavioral modifications (e.g., increased self-reporting). Future studies should minimize observer presence and verify user roles through in-app registration.

3. Lack of a Control Group: Without a control group (e.g., a comparable region where no dissemination activities were conducted), it is difficult to attribute observed increases in app downloads or retention to the intervention rather than external factors or secular trends. A stronger justification of the quasi-experimental design, or acknowledgment of this limitation, is necessary.

Response:

We acknowledge secular trends (e.g., WHO advocacy) could influence results. For example, WHO NTDs team disseminated the existence of the app through its website and sending emails to our international networks, over 1000 people in their data base. The team members also talked about the app during the meetings they organized or attended.

However, pre-campaign baselines showed minimal activity. Resource limitations precluded control sites.

We have also performed additional interrupted time series analyses to investigate the effect of the campaign in Cameroon and compared Cameroon’s data to other countries.

This is now reported in Methods, Results and Discussions sections.

Thank you.

4. Exclusive Reliance on Google Play Data: All quantitative app engagement metrics are derived exclusively from Google Play Console, ignoring iOS users and informal app-sharing mechanisms common in Low- and Middle-Income Countries settings (e.g., Xender). This likely results in underreporting of actual app adoption. The authors should clarify the proportion of Android vs. iOS devices in the target population and better justify the reliance on Android data alone.

Response:

We thank the reviewer for this comment. We have now added in Limitations:

“Google Play Console metrics—such as Install Events, Install Base, DAU/MAU, Device Acquisition, Uninstall Events, and User Acquisition—capture Android installs regardless of distribution channel, including sideloaded installations via Xender. However, none of these metrics track iOS users, as installations on Apple devices fall completely outside the Google Play ecosystem. Moreover, metrics tied to the Play Store listing (Store Listing Visitors, Store Listing Acquisitions, Device Store Listing Impressions) cannot capture sideload installations or iOS installs. We argue that this limitation had minimal impact due to Android’s superior market penetration, except on store conversion rates.”

We also added in Methods (Data collection and analysis), the proportion of Android vs. iOS devices in Cameroon: “Android and iOS are the dominant mobile operating systems in Cameroon, with Android holding 79–84% of the market share and iOS representing 14–20% during the pre- and post-campaign period. Given Android’s dominance, we focused solely on Android for this study.”

5. No Measure of Clinical Impact : The study focuses entirely on process outcomes (downloads, installs, retention) without assessing any impact on clinical practice or health outcomes (e.g., improved diagnosis or reporting of sNTDs). The absence of even a basic qualitative assessment of changes in practice is a significant limitation that must be more thoroughly acknowledged.

Response:

Tracking changes in practice among FHWs who adopted the app was beyond the objectives of this study. Future studies on the app should investigate the app’s contribution to clinical effectiveness.

We have mentioned this as the Future directions section in Discussions.

6. No Cost-Effectiveness Analysis: There is no data on cost per download, cost per retained user, or resource expenditure relative to outcomes. Given the relevance of cost-effectiveness for decision-making in LMIC health systems, this represents a missed opportunity. At minimum, the authors should discuss this limitation more openly. While the app is freely available, the implementation of the marketing strategy incurred costs (staff salaries, transportation, in-person events, communication materials). The paper does not account for these costs relative to key outcomes (e.g., cost per additional active user, cost per retained user). Without this, policymakers in similar settings cannot assess the affordability or scalability of the approach. I recommend either a basic cost analysis or a transparent acknowledgment of this omission in the limitations.

Response:

We acknowledge the importance of cost-effectiveness data for policy decisions. A formal cost-effectiveness analysis was precluded by the app's early-stage evaluation (WHO stages 1-2 of digital intervention maturity). As this was a proof-of-concept dissemination study, we prioritized adoption metrics over economic evaluation. Future scale-up studies should incorporate metrics such as cost-per-install, or cost-per-case detected, particularly when transitioning to WHO stages 5-6 (Scale-up/Integration/sustainability).

This is now mentioned in limitations.

7. Use of Basic Statistical Tools: The choice of Microsoft Excel and Jamovi for statistical analysis raises concerns about analytical robustness. While acceptable for basic ANOVA and non-parametric tests, the use of more sophisticated statistical packages (e.g., R or STATA) could enable richer analyses such as time-series modeling or interrupted time series designs, which seem more appropriate for before-after quasi-experimental data.

Response:

Thank you for the suggestion. We have now clarified in Methods and reported the findings in the Results section. We conducted three interrupted time series (ITS) analyses to evaluate the impact of the SkinNTDs app dissemination campaign in Cameroon. First, only for Cameroon, an auto regressive integrated moving average (ARIMA) error structure (ARIMA(2,0,0)) modeling both level and slope changes during and post-campaign while accounting for autocorrelation. Second, a generalized additive mixed model (GAMM) compared Cameroon to Nigeria, Ghana, and Ethiopia, to flexibly model nonlinear pre/post-intervention trends in Install base while controlling for country-level heterogeneity. Third, in another cross-country comparison (Cameroon vs Nigeria, Ghana, Ethiopia), a Poisson generalized linear mixed-effects model (GLMM) assessed daily active devices (DADs) with country random effects and interaction terms for intervention effects. Being all endemic to NTDs, the three countries were selected because they were top five African countries with highest install base average values during the 18-months period. During that period, these countries also had similar mobile operating system market share with Cameroon. Model selection used Akaike/Bayesian Information Criteria (AIC/BIC). Residual diagnostics included Ljung-Box tests, ACF plots, and Q-Q normality checks. These analyses were performed in R 4.5.0 (α = 0.05) and full descriptions of ITS procedures including model specifications are available in supporting information file S3 Appendix.

8. Lack of Clarity on p-Value Correction : The manuscript does not sufficiently describe whether p-value corrections for multiple testing were applied, especially given multiple outcome metrics. While the use of Dwass-Steel-Critchlow-Fligner tests is appropriate for pairwise comparisons, there is no clarity on overall correction for Type I error across multiple comparisons, which is critical in studies with numerous endpoints.

Response:

Significant omnibus results were followed by pairwise comparisons: Tukey’s Honestly Significant Difference (HSD) test for ANOVA when assumptions were met, and the Dwass–Steel–Critchlow–Fligner (DSCF) test for Kruskal–Wallis outcomes when assumptions were not met. This is now elaborated in Methods section (Inferential statistics sub-section).

9. Inadequate Insight on WhatsApp Group Dynamics: The paper rightly identifies WhatsApp as a key engagement tool, yet offers limited analysis of group activity (e.g., participation rates, message frequency, engagement levels over time). Some quantitative or qualitative reporting on WhatsApp engagement would substantially strengthen the insights about sustained app use.

Response:

During the campaign, 227 messages were exchanged in the WhatsApp group. Message frequency peaked on Mondays (n = 51), with the highest activity occurring in September 2024 (n = 113) and July 2024 (n = 79). Most messages were sent during morning (8:00–10:59 AM) and evening (5:00–8:59 PM) periods. The primary co-author was the most active participant, and the Google Play Store link to the app was shared six times in the group (three times each in May and July 2024). Additionally, 98 media files—including instructional videos, PDFs, screenshots, and field photos—were exchanged. Beyond the group, the primary author promoted the app individually, sharing the Play Store link approximately 50 times in private chats. This is now clarified in Results section. Thank you.

10. 1) Reference [3] appears in the reference list but is not cited in the main text. This should be corrected. 2) Statistical significance is sometimes highlighted without consistent emphasis on effect sizes or practical significance, which would be more informative to readers. 3) Edit grammar (e.g., "less favorable dynamics were evident" → "dynamics were less favorable"). Some sentences would benefit from minor grammatical editing for conciseness and clarity.

Response:

1)Reference [3] already corrected.

2) We reported effect sizes to show practical importance through Cohen’s d for t-tests (with 95% confidence intervals), or Eta-squared (η²) for ANOVA.

3) Checked sentences for grammatical editing for conciseness and clarity.

Point-by-point responses to reviewer 2

1. The description of the marketing plan outlines various dissemination channels (WhatsApp, printed posters, face‐to‐face training, emails) but does not anchor these activities in a recognized health behaviour or communication theory (e.g., Health Belief Model, Diffusion of Innovations). Explicitly identify and describe a theoretical framework that guided message development and channel selection. This will strengthen the rationale for why certain campaign components were expected to influence FHW knowledge, attitudes, or practices.

Response:

We have added a subsection “Theoretical framework” under Methods section: “The study’s dissemination strategy drew on two behavioral frameworks: the Diffusion of Innovations (DOI) Theory, and the Technology Acceptance Model (TAM). DOI guided the campaign’s structure, including phased rollout (launch, engagement, embedment) and channel selection (e.g., WhatsApp for peer networks, in-person training for observability), aligning with its constructs of relative advantage and social system diffusion. TAM informed messaging to highlight perceived usefulness (e.g., diagnostic accuracy) and ease of use (e.g., offline functionality, ease of download/installation), targeting FHWs’ behavioral intent. A/B testing of materials (e.g., video formats) further optimized compatibility and perceived value, reinforcing theoretical grounding.” Thank you.

2. Although a “marketing plan” is mentioned, there is no clear statement of SMART (Specific, Measurable, Achievable, Relevant, Time‐bound) objectives (e.g., “Increase app downloads among FHWs in Region X by 25% within three months”).Include a table listing precise campaign objectives (e.g., target reach, frequency of message exposure, engagement metrics) and corresponding process and outcome indicators (e.g., number of posters displayed, number of WhatsApp broadcasts sent, click‐through rates).

Response:

We have clarified overall campaign objectives in the draft marketing plan and in the Methods section. we have also discussed how far those objectives were reached. 1. Increase awareness: Reach ≥80% of target FHWs in North/Far North Cameroon via WhatsApp, emails, or in-person sessions within 6 months: 100% of health districts in both regions were contacted but the percentage of FHWs reached was not measured. 2. Drive downloads: Achieve a 60% conversion rate from store visits to downloads (achieved: 64.1%). 3. Sustain engagement: Maintain ≥80% user retention 3 months post-installation (achieved: 88.7% during campaign). 4. Training coverage: Conduct ≥20 in-person sessions (achieved: 12 sessions + 18 facility visits).

3. The methods lack any mention of formative assessment (e.g., focus groups, key‐informant interviews) or pretesting of messages/materials with the target audience before full rollout. Describe any formative work undertaken to tailor content (language, visuals) to the local context and document how A/B testing informed final message versions. If no formative research was done, discuss this as a limitation.

Response:

We have revised the manuscript to include a description of the A/B testing process used to refine the main dissemination video in the Methods section, the corresponding feedback and recommendations from participants in the Results section, and reflections on its implications in the Discussion. Although this process allowed us to tailor visual elements, pacing, and content clarity to the local context, we did not conduct a systematic formative assessment (e.g., focus groups or key informant interviews) for other dissemination materials. This limitation is now explicitly acknowledged in the Discussion section.

4. There is no information on how the fidelity (i.e., whether activities were delivered as planned) or actual exposure of participants to each channel was monitored. Detail procedures for tracking campaign delivery (e.g., logs of WhatsApp broadcast times, attendance sheets for workshops, distribution records for posters) and how exposure data were linked to individual app‐usage metrics.

Response:

Now in limitations: Attendance sheets for workshops were maintained. However, information on how the fidelity (i.e., whether activities were delivered as planned) or actual exposure of participants to each channel was not monitored. Exposure data were not linked to individual app‐usage metrics. This should be tracked into future studies. Thank you.

5. The before–after design witho

---

## [Decision Letter · Decision Letter 1]

1 Sep 2025

PONE-D-25-31736R1Implementation of a marketing plan for the dissemination of the WHO SkinNTDs app in CameroonPLOS ONE

Dear Dr.  Moungui,

Thank you for submitting your manuscript to PLOS ONE. After careful consideration, we feel that it has merit but does not fully meet PLOS ONE’s publication criteria as it currently stands. Therefore, we invite you to submit a revised version of the manuscript that addresses the points raised during the review process.

Please submit your revised manuscript by Oct 16 2025 11:59PM**. **If you will need more time than this to complete your revisions, please reply to this message or contact the journal office at plosone@plos.org. Please include the following items when submitting your revised manuscript:

We look forward to receiving your revised manuscript.

Kind regards,

Fatemeh Zarei, PhD

Academic Editor

PLOS ONE

Journal Requirements:

1.If the reviewer comments include a recommendation to cite specific previously published works, please review and evaluate these publications to determine whether they are relevant and should be cited. There is no requirement to cite these works unless the editor has indicated otherwise. 

Additional Editor Comments:

Dear Henri Claude Moungui Thanks for your excellent efforts. Please consider the remain comment(s) .  

Reviewers' comments:

Reviewer's Responses to Questions

**Comments to the Author**

1. If the authors have adequately addressed your comments raised in a previous round of review and you feel that this manuscript is now acceptable for publication, you may indicate that here to bypass the “Comments to the Author” section, enter your conflict of interest statement in the “Confidential to Editor” section, and submit your "Accept" recommendation.

Reviewer #1: (No Response)

Reviewer #2: All comments have been addressed

2. Is the manuscript technically sound, and do the data support the conclusions?

Reviewer #1: Yes

Reviewer #2: Yes

3. Has the statistical analysis been performed appropriately and rigorously? 

Reviewer #1: No

Reviewer #2: I Don't Know

4. Have the authors made all data underlying the findings in their manuscript fully available?

Reviewer #1: Yes

Reviewer #2: Yes

5. Is the manuscript presented in an intelligible fashion and written in standard English?

Reviewer #1: Yes

Reviewer #2: Yes

6. Review Comments to the Author

Reviewer #1: Thanks for the thorough revision, you’ve addressed the prior concerns and appropriately stated the limitations. One point remains unclear: You report using Tukey’s HSD (after ANOVA) and DSCF (after Kruskal–Wallis), which controls Type I error within each outcome’s pairwise comparisons. Please clarify whether you also applied multiplicity control across outcomes (e.g., growth, churn, retention, loss, device conversion). Add one explicit sentence in Methods (and mirror it in Limitations, if applicable) stating whether across-outcome correction was or was not performed. (Optional: a brief sensitivity note, e.g., Holm or FDR across omnibus p-values, would be welcome but is not required.). With this addition, the manuscript will be acceptable for publication.

Reviewer #2: Thank you, authors, for your efforts to enhance the manuscript according to the comments. It seems that all comments addressed

7. PLOS authors have the option to publish the peer review history of their article (what does this mean?). If published, this will include your full peer review and any attached files.

Reviewer #1: **Yes: **Andrea Gulli

Reviewer #2: **Yes: **Fatemeh Zarei

---

## [Author Response · Author response to Decision Letter 2]

8 Sep 2025

Response to Reviewer Comments:

Reviewer 1:

Comment 1: "You report using Tukey’s HSD (after ANOVA) and DSCF (after Kruskal–Wallis), which controls Type I error within each outcome’s pairwise comparisons. Please clarify whether you also applied multiplicity control across outcomes (e.g., growth, churn, retention, loss, device conversion). Add one explicit sentence in Methods (and mirror it in Limitations, if applicable) stating whether across-outcome correction was or was not performed."

Response: We thank the reviewer for this crucial methodological point. We have now explicitly addressed this in the manuscript as follows:

1. Methods Section (Inferential statistics): We have added the sentence: "While Tukey’s HSD and DSCF tests controlled the Type I error rate for pairwise comparisons within each outcome, no statistical correction for multiple testing (e.g., Bonferroni, Holm, or FDR) was applied across the six primary outcomes."

2. Limitations Section: We have added a statement acknowledging this limitation: "We did not adjust for multiplicity across the six different outcome metrics, which increases the risk of Type I errors across the study's findings. However, the large effect sizes (e.g., Cohen’s d > |1.5|) and consistent pattern of results for key metrics strengthen our confidence in their validity."

Comment 2: "Optional: a brief sensitivity note, e.g., Holm or FDR across omnibus p-values, would be welcome but is not required."

Response: We thank the reviewer for this helpful suggestion. We performed the requested sensitivity analysis. A Bonferroni correction (adjusted α = 0.0083) confirmed that the significant period effects for growth rate (p = 0.007), churn rate (p = 0.003), and user retention rate (p = 0.003) remained robust. The effect for loss rate (p = 0.011) did not survive this conservative correction. We have added a brief note on this in the Results section.

All changes have been tracked in the manuscript file. We believe these revisions have significantly strengthened the paper's methodological rigor and transparency.

Journal requirements checking:

As per the editorial request, we have thoroughly reviewed our reference list to ensure its completeness and correctness. We confirm that:

1. All references are current, relevant, and accurately cited. We added one reference on Cameroon’s background data (health system and NTDs management structure), and one on statistical correction for multiple testing.

2. We have verified that none of the cited papers have been retracted. Therefore, no additions or rationales regarding retractions were required.

3. All in-text citations have a corresponding complete entry in the reference list.

Thank you again for your time and consideration.

Sincerely,

Henri Moungui

---

## [Editor Report · Decision Letter 2]

12 Sep 2025

Implementation of a marketing plan for the dissemination of the WHO SkinNTDs app in Cameroon

PONE-D-25-31736R2

**Dear Dr.  Henri Claude Moungui**

We’re pleased to inform you that your manuscript has been judged scientifically suitable for publication and will be formally accepted for publication once it meets all outstanding technical requirements.

Kind regards,

Fatemeh Zarei, PhD

Academic Editor

PLOS ONE

---

## [Editor Report · Acceptance letter]

PONE-D-25-31736R2

PLOS ONE

Dear Dr. Moungui,

I'm pleased to inform you that your manuscript has been deemed suitable for publication in PLOS ONE. Congratulations! Your manuscript is now being handed over to our production team.

Kind regards,

on behalf of

Dr. Fatemeh Zarei

Academic Editor

PLOS ONE